# Make-or-break prime editing for genome engineering in *Streptococcus pneumoniae*

Monica Rengifo-Gonzalez[1,3], Maria-Vittoria Mazzuoli [1,3], Axel B. Janssen [1,3], Anne-Stéphanie Rueff[1], Jessica Burnier [1], Xue Liu [2] ✉ & Jan-Willem Veening [1] ✉

CRISPR-Cas9 has revolutionized genome engineering by allowing precise introductions of DNA double-strand breaks (DSBs). However, genome engineering in bacteria is still a complex, multi-step process requiring a donor DNA template for repair of DSBs. Prime editing circumvents this need as the repair template is indirectly provided within the prime editing guide RNA (pegRNA). Here, we developed make-or-break Prime Editing (mbPE) that allows for precise and effective genetic engineering in the opportunistic human pathogen *Streptococcus pneumoniae*. In contrast to traditional prime editing in which a nicking Cas9 is employed, mbPE harnesses wild type Cas9 in combination with a pegRNA that destroys the seed region or protospacer adjacent motif. Since most bacteria poorly perform template-independent end joining, correctly genome-edited clones are selectively enriched during mbPE. We show that mbPE is RecA-independent and can be used to introduce point mutations, deletions and targeted insertions, including protein tags such as a split luciferase, at selection efficiencies of over 93%. mbPE enables sequential genome editing, is scalable, and can be used to generate pools of mutants in a high-throughput manner. The mbPE system and pegRNA design guidelines described here will ameliorate future bacterial genome editing endeavors.

The use of CRISPR/Cas9 technology has facilitated genetic engineering for a wide range of bacteria, including difficult or laborious to transform species such as *Staphylococcus aureus*, *Clostridium* spp. and *Streptomyces* spp[1–5]. The basis for CRISPR/Cas9-based bacterial genome editing is the selective enrichment of transformants by targeting *Streptococcus pyogenes* Cas9 to the genetic location of interest by a single guide RNA (sgRNA) followed by production of a double-stranded break (DSB)[6]. Thus, cells in which the donor template DNA has not been incorporated through homologous recombination, will be targeted thereby enriching for edited clones[1,7,8]. This is particularly efficient in bacteria, as most strains lack (or poorly perform) template-independent end joining mechanisms such as non-homologous end joining (NHEJ) or alternative end joining[2,9,10]. Thus, only clones edited at the DSB and thereby disrupting the protospacer adjacent motif (PAM) or sgRNA seed region will survive. The major drawback of CRISPR/Cas9-based genome editing in bacteria is that several steps are involved such as the cloning of an sgRNA and the production or cloning of a template DNA[1,11].

Recently, prime editors such as PE2 were described in which a Cas9 nickase is fused to an engineered reverse transcriptase[7,12]. PE2 has the advantage that it does not introduce DSBs, which, in eukaryotes typically leads to error-prone end-joining via NHEJ and has the risk of inducing chromosomal rearrangements[7]. In addition, prime editing uses a pegRNA for both targeting of the PE2 complex as well as a

[1]Department of Fundamental Microbiology, Faculty of Biology and Medicine, University of Lausanne, Biophore Building, CH-, Lausanne, Switzerland. [2]Department of Pathogen Biology, Base for International Science and Technology Cooperation: Carson Cancer Stem Cell Vaccines R&D Center, International Cancer Center, Shenzhen University Medical School, Shenzhen, Guangdong, China. [3]These authors contributed equally: Monica Rengifo-Gonzalez, Maria-Vittoria Mazzuoli, Axel B. Janssen. ✉e-mail: xueliu@szu.edu.cn; Jan-Willem.Veening@unil.ch

template for DNA repair in which the desired mutation is encoded. The pegRNA contains a spacer sequence to direct targeting of the prime editor to the PAM site, a primer binding site (PBS) that facilitates the hybridization of the 3′-end of the nicked DNA strand to the pegRNA and includes a reverse transcriptase template (RTT) that carries the intended genetic modification[12]. Because of these advantages, prime editing has rapidly gained traction and has been successfully applied to engineer the genomes in various eukaryotes including plants[13–16]. However, prime editing has not yet been widely applied for bacterial genome editing, probably because of limitations in the selection efficiency during the introduction of PE2-mediated mutations or due to the requirements of specific genetic backgrounds[17,18]. For instance, by mutating three 3′→5′ DNA exonucleases, PE efficiencies in *Escherichia coli* were improved 100-fold[18].

Here, we describe make-or-break prime editing (mbPE) as an efficient genome editing method in the opportunistic human pathogen *Streptococcus pneumoniae*. mbPE is a nuclease-based prime editing system that uses wild type *S. pyogenes* Cas9 fused to a bacterial codon-optimized reverse transcriptase under inducible control. Similar nuclease-based PE strategies have been employed in eukaryotic systems[19–21], but not yet in bacteria. Using mbPE, we were able to make point mutations, deletions over 100 bps as well as insertions up to 52 bps in a targeted fashion at high selection efficiencies. Using a pooled high-throughput approach, we established mbPE design rules and show that an optimal pegRNA, to insert a single adenine base, consists of a PBS of 16 nt and a RTT of 15 nt. We provide pegRNA vectors with different antibiotic selection markers allowing for sequential pneumococcal genome edits. The mbPE system developed here will be of great value to the pneumococcal research community and may function as a starting point for make-or-break genome editing approaches in other bacteria.

## Results

### Prime editing in *S. pneumoniae*

Several Cas9-based tools have been developed for *S. pneumoniae*[22–26], but prime editing has not been implemented yet. To establish a practical prime editing system for *S. pneumoniae*, we first focused on the prime editing tool PE2. PE2 is a fusion of the *S. pyogenes* Cas9[H840A] nickase domain and an engineered Moloney murine leukemia virus reverse transcriptase domain (MMLVRT)[12]. Previously, we described tight inducible control by the addition of tetracycline-group molecules (including anhydrotetracycline (aTc)) of a catalytically inactive Cas9 (dCas9: D10A, H840A) expressed as a single copy from the non-essential chromosomal CEP locus ($P_{tet}$-*dcas9*) for CRISPR interference (CRISPRi) in *S. pneumoniae*[25]. In a first step to a functioning PE2 for *S. pneumoniae*, we mutated the alanine residue at position 10 of the $P_{tet}$-*dcas9* construct back to an aspartic acid to create $P_{tet}$-*cas9*[H840A]. The engineered MMLVRT domain and a linker domain from PE2 were codon optimized for optimal expression in low GC-rich bacteria such as *S. pneumoniae* and cloned to the C-terminus of the Cas9 nickase resulting in strain VL4199 ($P_{tet}$-*cas9*[H840A]-MMLVRT or $P_{tet}$-PE2[S,pn]) (Fig. 1A).

Next, a pegRNA construction and *S. pneumoniae* integration vector was designed in which the pegRNA is constitutively expressed from the constitutive P3 promoter including its +1 adenine transcription start site to ensure that all cloned pegRNAs are efficiently transcribed and do not contain undesirable uridines at the 5′ end[26–28]. For easy cloning, an *mCherry* cassette flanked by Esp3I (isoschizomer of BsmBI) sites is present allowing for rapid, oligo-based, Golden Gate cloning of pegRNAs (Fig. 1B). Sequence elements facilitating one-step library preparation for NGS were also introduced[29] resulting in pegRNA cloning vectors pVL4132, pVL4133 and pVL4393 (carrying 7 thymine nucleotides, a *S. pneumoniae* terminator or a *S. pneumoniae tRNA-asp-1* after the extension sequence, respectively). pegRNAs consist of four elements: 1) a 20 nt spacer sequence for targeting of the prime editor,

2) a scaffold sequence for binding to Cas9, 3) a primer binding site (PBS) that allows the 3′-end of the nicked DNA strand to hybridize to the pegRNA, and 4) a reverse transcriptase template (RTT) containing the desired edit[12]. Since bacterial transcription termination leaves nucleotides at the 3′-end of the pegRNA, which may interfere with efficient prime editing downstream, we cloned *S. pneumoniae tRNA-asp-1* directly following the pegRNA (Fig. 1B). This strategy has been previously used to produce multiple individual sgRNAs from a single polycistronic operon as the endogenous tRNA-processing system precisely cleaves both ends of the *tRNA-asp-1* precursor[30]. To test whether the PE2[S,pn] is functional, we designed several pegRNAs using PrimeDesign[31] to introduce either a premature stop codon in the firefly luciferase gene (*luc*) expressed in pneumococcal strain VL4200, or to introduce a repairing mutation in strain VL4297 in which *luc* has a premature stop codon (Fig. 1C). Interestingly, induction of PE2[S,pn] together with a targeting pegRNA did not result in significant growth delay, suggesting that *S. pneumoniae* can tolerate and readily repair ssDNA breaks (Supplementary Fig. 1A). As shown in Fig. 1D, ~2% of clones in which PE2[S,pn] was induced demonstrated the desired bioluminescence phenotype. Sanger sequencing of several individual clones confirmed the *luc* expression observations, demonstrating that bioluminescence is a good proxy for DNA editing (Supplementary Fig. 1B, C).

Recently, it was shown that a structured 3′-end of a pegRNA increases stability and improves editing efficiency in human cells[32]. In addition, a functional mismatch repair system was shown to negatively affect editing efficiencies[33,34]. To test whether the addition of a structured RNA motif at the 3′-end would also improve PE2 editing efficiency in *S. pneumoniae*, we replaced the tRNA sequence for the structured and highly efficient *S. pneumoniae rpsT* transcriptional terminator[35] (SpTer) or a stretch of 7 thymine nucleotides (7 Ts) (Fig. 1E). As shown in Fig. 1D, luminescence data of both pegRNAs suggests improved efficiency of inserting a single A in *luc* (~3%) through PE2[S,pn]-mediated editing, compared to the tRNA-Asp1 system. Sanger sequencing of several bioluminescent-negative clones confirmed the microtiter plate assay results (Supplementary Fig. 1). Repairing an internal stop codon (TAA to GCT) in *luc* was ~3 times more efficient with the SpTer terminator compared to the tRNA-Asp1 construct (Fig. 1D). Deleting the pneumococcal mismatch repair gene *hexA*[36] further enhanced PE2[S,pn] editing efficiency in the constructs with either the 7 T or SpTer terminators (Fig. 1F). These results demonstrate that prime editing using PE2[S,pn] in combination with a structured pegRNA works in *S. pneumoniae* and can be used to incorporate insertions and mutations, albeit at a relatively low selection efficiency.

### Make-or-break prime editing (mbPE) in *S. pneumoniae*

Since prime editing depends on successful hybridization of the 3′-end of the nicked DNA to the pegRNA[12], we wondered if prime editing would also be functional when, instead of a nick of the PAM-containing strand, a DSB is introduced. Recent work in human HEK293T cells showed efficient prime editing utilizing Cas9-induced DSBs that largely depend on NHEJ[19–21]. To test if DSB-driven prime editing (here after called make-or-break prime editing: mbPE) would also work in bacteria lacking NHEJ, we first mutated the alanine residue at position 840 within the Cas9 nickase moiety of PE2[S,pn] back to a histidine. This led to full restoration of wild type Cas9, resulting in strain VL5149 ($P_{tet}$-*cas9*-MMLVRT or mbPE[S,pn]) (Fig. 2A, B). This construct was combined with *luc* expression (strain VL5155) and the same set of pegRNAs containing the SpTer or 7 Ts as previously used to verify the efficiency of the nickase-based PE2[S,pn] (Fig. 1E). All pegRNAs were designed to either disrupt the PAM or the seed region within *luc*. Thus, only correctly edited bacteria are expected to survive the expression of mbPE[S,pn], as unedited repairs will lead to continued DSBs, hence the name make-or-break prime editing.

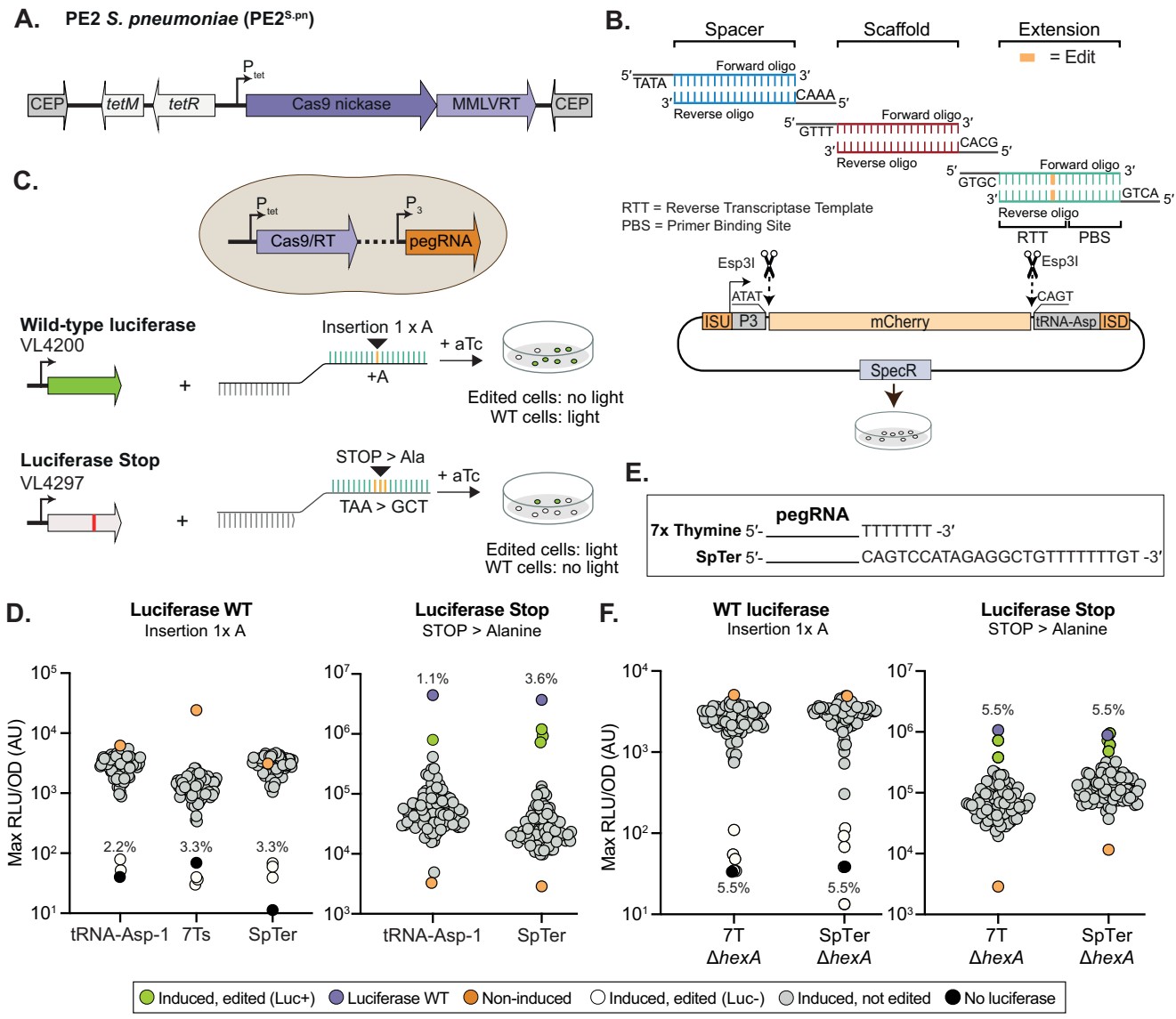

**Fig. 1 | Prime editing 2 (PE2) in *S. pneumoniae*. A** Schematic overview of PE2^*S.pn*. **B** Schematic overview of the general cloning strategy of pegRNAs (shown is the *tRNA-asp-1* containing vector pVL4393 as backbone). **C** Editing strategy used to test efficiency of PE2^*S.pn* in pneumococcal strains expressing either a functional luciferase (VL4200) or a *luc* gene interrupted by a premature stop codon (VL4297). Single colonies are picked and grown in microtiter plates in the presence of luciferase substrate and bioluminescence is recorded. **D** Induction of PE2^*S.pn* by aTc in combination with expression of a pegRNA without 3′ leader, by processing of the *tRNA-asp-1*, leads to accurate gene editing in ~2% of *S. pneumoniae* cells. Induction of PE2^*S.pn* in combination with 3′ extended pegRNAs provides an editing efficiency of ~3%. Each dot represents a randomly picked clone grown in C + Y medium containing luciferin substrate. The maximal bioluminescence (relative light units; RLU) reached, normalized by the optical density (OD), of each clone is shown (AU, arbitrary units). RLU/OD values were normalized against the baseline level of medium luminescence obtained for each experiment. A typical experiment is shown (at least 3 biological replicates, with *n* = 90). **E** Schematic overview of pegRNAs containing the *S. pneumoniae rpsT* terminator (SpTer) or a stretch of 7 Ts. **F** Prime editing is slightly enhanced in the absence of mismatch repair (Δ*hexA*) in *S. pneumoniae* (*n* = 90). Clones with either an RLU/OD value below 2 × 10² (left), or an RLU/OD value above 7 × 10⁶ (right), were considered to be edited (as also validated by Sanger sequencing, Supplementary Fig. 1).

Induction of mbPE^*S.pn* with aTc, in the presence of a targeting pegRNA, caused severe growth defects indicating that DSBs were formed (Supplementary Fig. 2A). The system is tightly inducible as reported previously[37], as we observed a similar number of colonies compared to a strain that contained a non-targeting pegRNA plated in the absence of aTc (Supplementary Fig. 2B), suggesting that DSBs are only generated in the presence of inducer. Strikingly, surviving colonies (~0.1%, depending on the pegRNA and aTc concentration), predominantly displayed the correct luciferase phenotype (93% of selected colonies) (Fig. 2C). Sanger sequencing demonstrated that the precise edit was made as encoded by the pegRNA template (Supplementary Fig. 2C–F). Colonies that did not display the correct

*luc*-edit either had suppressor mutations in the Cas9 domain, or deletions in the $P_{tet}$ promoter driving the expression of mbPE^*S.pn* (Supplementary Fig. 2G). When comparing an sgRNA targeting *luc* with a pegRNA encoding the same spacer sequence, we observed an increase in recovered colonies when the pegRNA was used, in line with the editing efficiencies observed with PE2 (Supplementary Fig. 2B). Note that the reverse transcriptase domain was essential for prime editing since we did not observe any correctly edited clones when only the Cas9 domain was expressed (Fig. 2D). PacBio HiFi long read whole genome sequencing of four edited clones did not reveal signs of off-target editing nor any genomic rearrangements (Supplementary Fig. 3B, C) (SRA accession number PRJNA1062582).

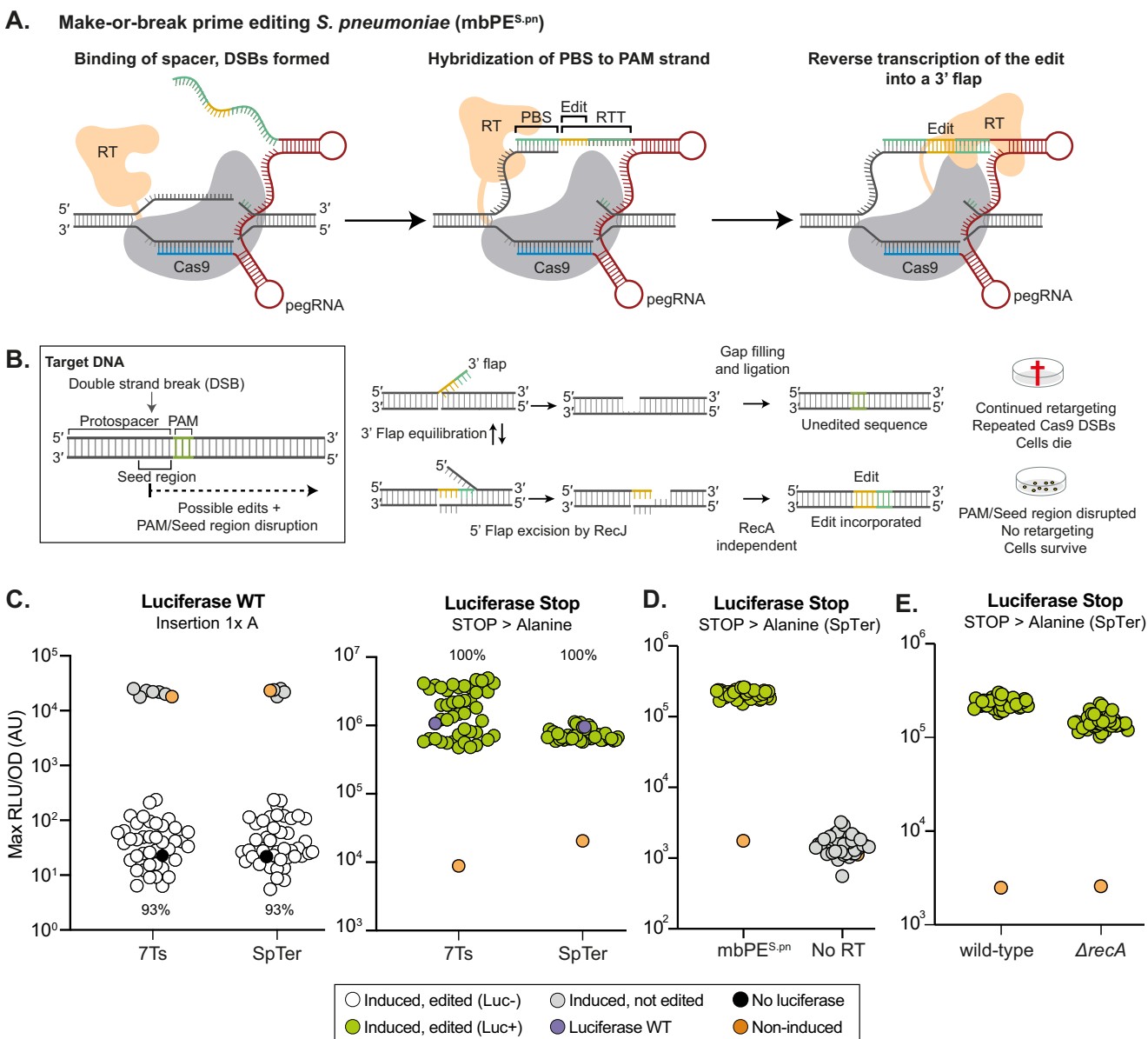

**Fig. 2 | Make-or-break prime editing selectively enriches correctly edited clones. A** Schematic overview of mbPE$^{S.pn}$. Binding of the spacer sequence to the target region leads to formation of DSBs by the Cas9 domain. The PBS can hybridize to its complementary sequence. The reverse transcriptase (RT) domain of mbPE$^{S.pn}$ subsequently synthesizes cDNA using the RTT as template. **B** Schematic depicting the mechanism leading to selective survival of edited clones. The hybridization of the 3′-flap containing the edit is in equilibrium with the unedited strand. Repair with the edited region will prevent repeated DSBs by Cas9. **C** Induction of mbPE$^{S.pn}$ by aTc in combination with expression of a structured pegRNA leads to accurate gene editing. ~93% of surviving *S. pneumoniae* cells after induction are accurately edited

($n = 47$ each). Cells were grown in C + Y medium containing luciferin substrate. The maximal bioluminescence reached of each clone is shown (RLU: relative light units. OD: optical density). RLU/OD values were normalized against the baseline level of medium luminescence obtained for each experiment. In these experiments, clones with either an RLU/OD value below $2 \times 10^3$ (left), or an RLU/OD value above $2 \times 10^5$ (right), were considered to be edited (as also validated by Sanger sequencing). **D** In the absence of reverse transcriptase (strain VL7598, P$_{tet}$-*cas9*), no induced clones were edited, as indicated by the lack of bioluminescence ($n = 44$). **E** mbPE$^{S.pn}$ in *recA*-deficient strain shows that RecA-dependent homologous recombination is not required for editing ($n = 42$).

## mbPE does not require RecA-dependent homologous recombination

In Fig. 2D we established that reverse transcription of the pegRNA is required for correct genome editing. DNA insertion in bacteria can occur by specific bacteriophage recombinases but typically is established by RecA-dependent homologous recombination[23,38]. However, RecA typically requires long homology regions. To test whether RecA is indeed not required for make-or-break prime editing in *S. pneumoniae*, in which only short regions of homology are presented within the pegRNA, we replaced *recA* with an antibiotic

resistance cassette in strain VL7487 (mbPE$^{S.pn}$, pegRNA-luc+SpTer). Note that we were not able to construct this strain at 37 °C but could efficiently construct it at 30 °C. After inducing the mbPE system and picking 42 surviving colonies on aTc plates, all demonstrated the encoded *luc* repair in the *recA* strain (Fig. 2E). This data is in line with the original model for prime editing[12] in which a 5′→3′ ssDNA exonuclease (RecJ in *S. pneumoniae*) would excise the 5′ flap followed by gap filling by DNA polymerase and subsequent ligation by DNA ligase (Fig. 2B), a process that would not require RecA-dependent homologous recombination.

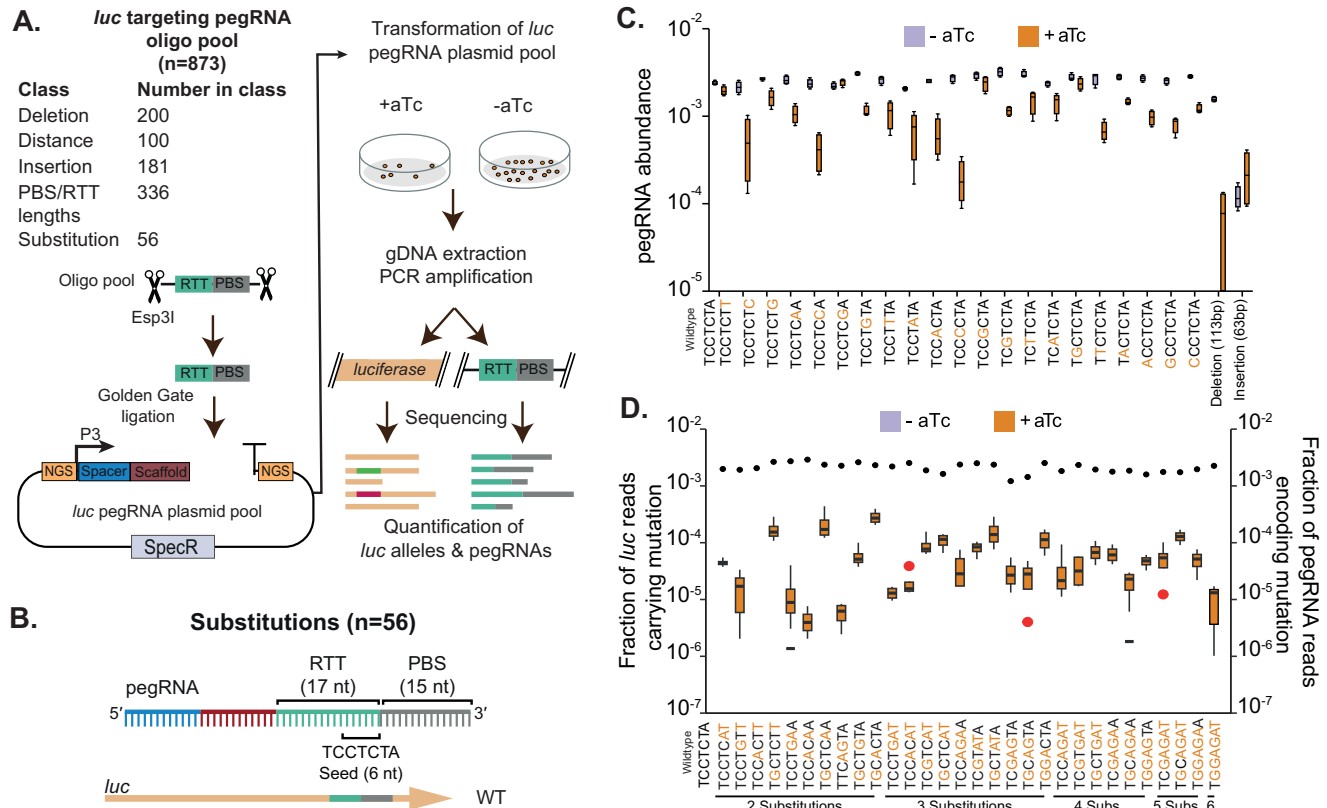

**Fig. 3 | mbPE^{S,pn} is highly scalable and efficient in making base pair substitutions. A** Cloning strategy to produce a library of pegRNAs targeting *luc* from a ssDNA oligo pool. The oligo pool was amplified by PCR as well as the backbone of pVL4134 that includes the *luc* spacer sequence and assembled by Golden Gate digestion/ligation followed by transformation to *S. pneumoniae* containing *luc* and mbPE2^{S,pn}. Chromosomal DNA was isolated from colonies grown in absence or presence of aTc (to induce mbPE2^{S,pn}) and PCR amplified using staggered primers followed by Illumina sequencing (see "Methods"). 2FAST2Q[63] was used to extract and count pegRNAs and *luc* alleles. **B** Schematic overview of the pegRNA design encoding for different base pair substitutions. (**C, D**) Single base pair substitutions and multiple base pair substitutions are installed with high selection efficiency. For mutation frequencies, results of four biological replicates are plotted in box-plots, with first quartile, median, and third quartile indicated. Whiskers indicate the maximum value, at most 1.5x IQR from the first or third quartile. Outliers are plotted in red. For pegRNA frequencies, the mean of four experiments is plotted. **C** pegRNA abundance of - aTc samples (control, purple) and + aTc samples (orange) are shown. For reference, values of a successful edit (113 bp deletion) and unsuccessful edit (63 bp insertion) are also plotted. **D** The fraction of *luc* reads carrying the desired mutation over the total reads in that sample is shown for induced (aTc treated, orange) and non-induced (-aTc; control) samples on the left Y-axis. The average fraction of pegRNA reads for the substitutions is shown as dots on the right Y-axis. Red dots indicate outliers in induced samples. Non-induced samples only have data indicated as lines in double mutant "TCCTGAA" and triple mutant "TGCAGAA", as only one non-zero datapoint was collected in those samples.

## Scalable pooled pegRNA library construction shows efficient installment of base pair substitutions

To explore the potential of mbPE^{S,pn}, we generated a pool of different pegRNAs targeting *luc* (Supplementary Data 1), followed by Golden Gate cloning (Fig. 3A). A total of 873 oligos were designed and synthesized as a pool, containing different PBS and RTT sequences to introduce various genome edits. The pegRNA pool was cloned into plasmid pVL4134 that already contains the *luc*-targeting spacer sequence. The resulting pool was subsequently used to transform strain VL5155 carrying *luc* and the aTc-inducible mbPE^{S,pn} (see "Methods"). The library was plated on agar lacking or containing 20 ng/ml aTc to induce mbPE^{S,pn}. Quadruplicate induced and uninduced experiments, each with a minimum of 40000 colonies, were pooled, and their pegRNA contents and edited *luc* region were sequenced (Fig. 3A). In the uninduced pool, all but 12 of the designed pegRNAs were present, and we did not observe some of the pegRNAs encoding the longest insertions. Strikingly, most pegRNAs were also present in the induced pool, albeit at varying abundances. By counting and comparing the number of reads coming of each pegRNA in both conditions, we could estimate a selection efficiency score for each pegRNA for a specific mutation (Fig. 3A). By also sequencing the *luc* gene, we confirmed that most of the designed edits were correctly made in the pool of mutants (see "Methods"). We note that while comparing the ratios of reads of pegRNAs is useful to assess whether a certain edit is in principle possible, particularly when the pegRNAs encode the same spacer and PBS, it does not quantitatively inform on editing efficiencies as selection efficiencies will be very different depending on the level by which the pegRNA-encoded mutation in the RTT affects targeting or editing. For instance, some pegRNAs might be less efficient at cutting, potentially due to improper folding, which in turn might enhance editing efficiency since the RT moiety has more time to produce the cDNA before a DSB is made.

First, we examined whether mbPE^{S,pn} was able to install various substitutions (a > t, a > c, a > g, t > a, t > c, t > g, c > a, c > t, c > g) of targets within a seed region. We designed pegRNAs that have the same 20 nt *luc* spacer and 15 nt PBS, but varying RTTs encoding the different substitutions (and the necessary *luc* seed disruptions) (Fig. 3B, and Supplementary Data 1). As shown in Fig. 3C, pegRNAs encoding all substitutions were present at high frequencies under induced conditions. Next, we wondered if mbPE^{S,pn} could introduce multiple substitutions simultaneously. Indeed, all tested substitutions, up to the 6 adjacent substitutions tested (CCTCTA > GGAGAT), were present in the induced library (Fig. 3D). For single base pair substitutions, selection efficiency was more efficient when the edit was made to destroy

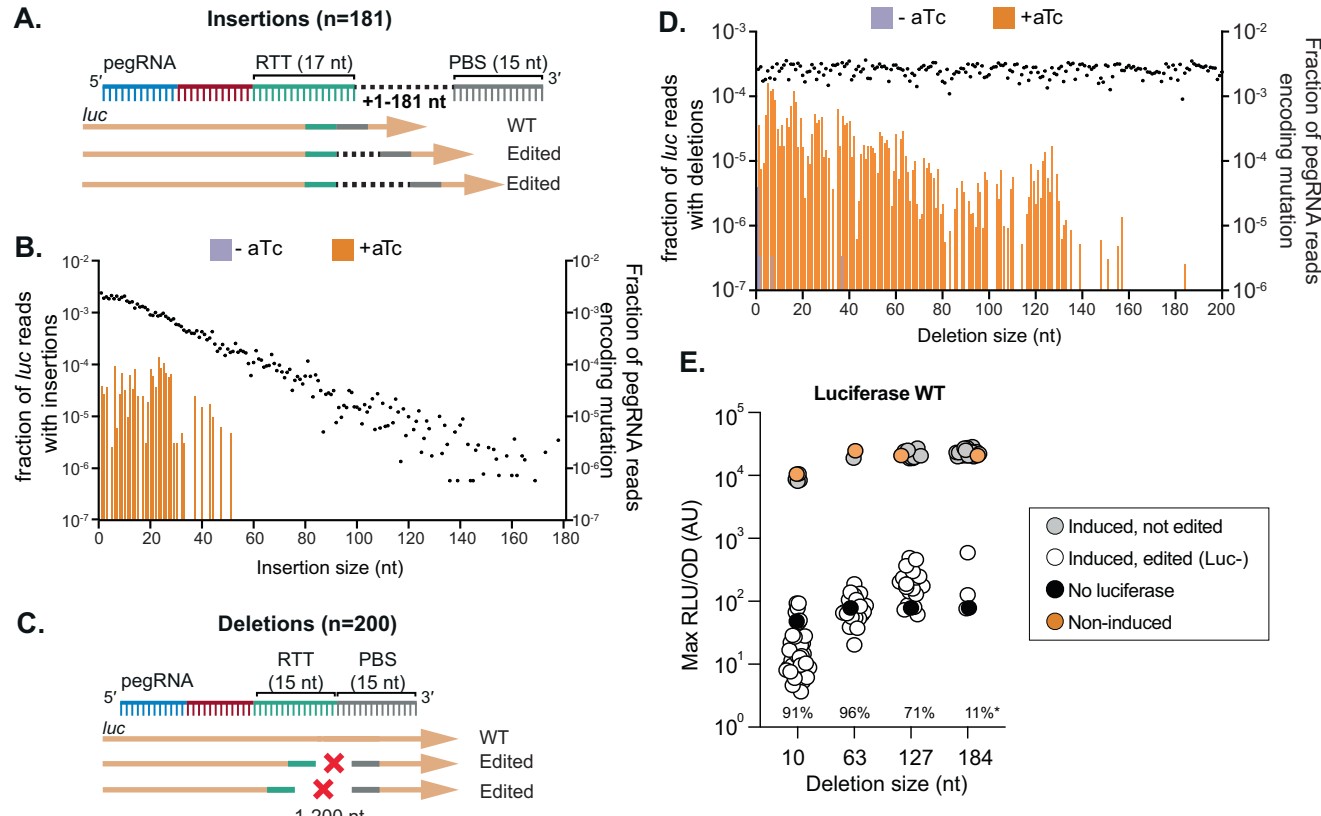

**Fig. 4 | mbPE for DNA insertions and genome deletions. A, C** Schematic overview of the approach used to test insertion and deletion efficiency in *luc* using mbPE. For pegRNAs encoding insertions, additional nucleotides were added between the RTT and PBS. For deletions, the RTT portion of the pegRNA was progressively moved within the *luc* gene, increasing deletion size, whilst the RTT remained constant. **B, D** Possibility of making insertions (**B**), and deletions (**D**) using mbPE. For each modification size, the fraction of *luc* reads carrying that mutation over the total reads in that sample, is displayed on the left axis with bars, whilst the fraction of

pegRNA reads from uninduced samples carrying the mutation is displayed with dots mapping to the right axis. Resulting means of four biological replicates are plotted. **E** Verification of the ability to make deletions in *luc*, through targeted deletion of 10, 63, 127, or 184 bp ($n = 45$, $n = 28$, $n = 28$). Clones with an RLU/OD value below $10^3$ were considered to be edited (as also validated by Sanger sequencing). The percentages indicate the fraction of clones that were phenotypically observed to be edited. Note that for the 184 bp deletion, no clone showed correct editing by Sanger sequencing (*).

the PAM and less efficient when the edit was distal from the PAM (Supplementary Fig. 4A).

## Efficient DNA insertion and deletion with mbPE

After establishing that mbPE$^{S,pn}$ is highly effective at selecting clones containing point mutations, we examined the capacity of mbPE$^{S,pn}$ for DNA insertions or deletions (Fig. 4A, C). To test DNA insertion efficiencies, we designed 181 different insertion pegRNAs targeting *luc* with a constant 20 nt spacer, a constant 15 nt PBS, a constant 17 nt RTT scaffold followed by an RTT insertion of varying length (between 1 and 181 nt) that would insert in the *luc* seed region thereby preventing Cas9 retargeting (Fig. 4A, and Supplementary Data 1). We analyzed pegRNA abundances of uninduced and induced libraries, as well as sequenced the *luc* genes in the pools. We observed insertions up to 52 bps, indicating that there is a maximum length that can be installed through mbPE in *S. pneumoniae* (Fig. 4B). Note that longer pegRNAs were less well represented in the insertion pegRNA pool likely because of reduced abundance due to uneven synthesis and/or cloning efficiencies (Fig. 4B). This could have influenced our ability to observe the mutations encoded in these larger pegRNAs.

A similar strategy was employed to assess deletion efficiency inside *luc*, and 200 deletion pegRNAs were designed each containing a constant 20 nt *luc* spacer and constant 15 nt PBS. The RTTs were 15 nt long, each shifted by one base pair relative to each other to create varying deletion sizes in *luc* whilst simultaneously disrupting the seed

region (Fig. 4C). We observed a decrease in editing efficiency as deletion length increased, with the longest deletion observed being 184 bp (Fig. 4D). Indeed, we could readily obtain deletions at high selection efficiency when reconstructing a set of deletion pegRNAs encoding the 10, 63 or 127 base pair deletion (ranging between 80–90% efficiency). However, only 3 out of 28 tested clones showed reduced luciferase activity with the 184 bp deletion pegRNA and Sanger sequencing indicated that these were not correctly edited at the 184 bp position (Fig. 4E).

## pegRNA design rules for optimal mbPE

Previous work in human cells has shown that PE2 gene editing efficiencies highly depend on RTT and PBS length[12]. To determine the optimal RTT and PBS length for mbPE$^{S,pn}$ to introduce a single adenine nucleotide insertion in *S. pneumoniae*, we designed 336 pegRNAs, each encoding the same 20-nucleotide *luc* spacer sequence but varying in RTT length (6-26 nucleotides) and in PBS length (5-20 nucleotides). Each RTT contained the insertion of a single adenine nucleotide to disrupt the *luc* PAM, thereby preventing retargeting of Cas9 (Fig. 5A). Differential enrichment analysis of pegRNA abundance with and without mbPE$^{S,pn}$ inducer showed relatively more surviving clones with a PBS longer than 16 nts, and an RTT longer than 11. The most enriched pegRNAs for inserting a single adenine in *luc* contained a combination of a RTT length of 14 or 15 with a PBS length of 16 or 17 nts (Fig. 5B). Together, these experiments provide guidelines for optimal pegRNA

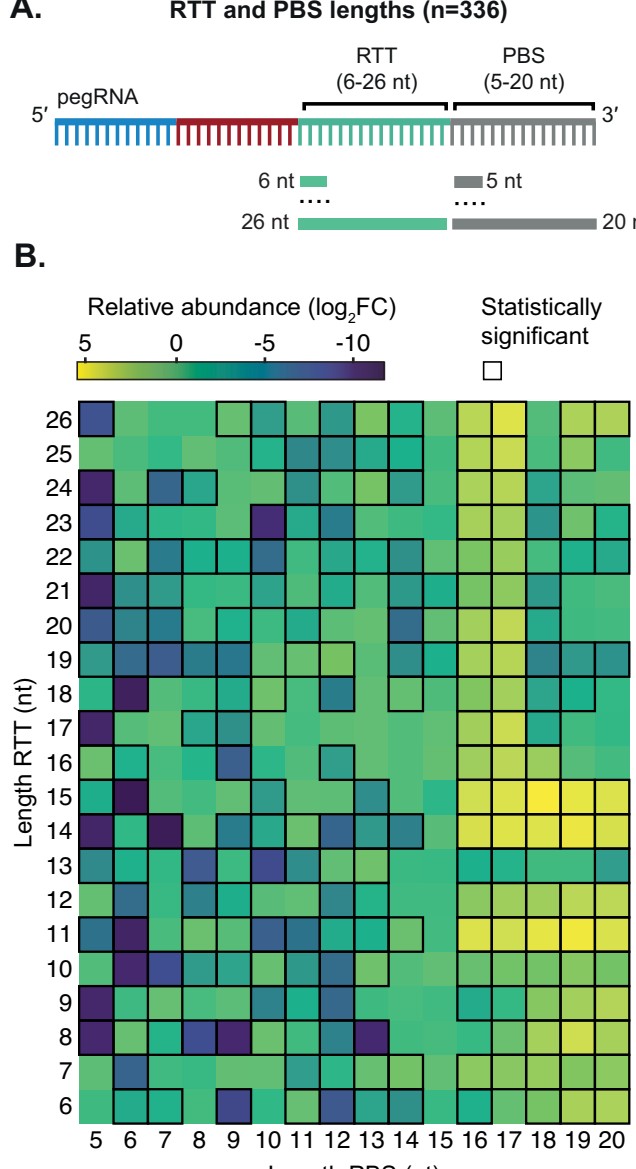

**A.**

**RTT and PBS lengths (n=336)**

**B.**

**Fig. 5 | pegRNA design rules for mbPE^{S,pn}. A** Schematic overview of the approach used to test mbPE^{S,pn} efficiency at inserting a single adenine in *luc*. **B** Combinations of RTT and PBS with statistically significant differences in pegRNA abundance between uninduced vs induced are outlined. Overrepresented combinations in induced samples show higher log2 fold changes (log$_2$FC) and are colored bright yellow. Most overrepresented pegRNAs of the induced library compared to uninduced suggest that a 16 or 17 nts for the PBS and 14 or 15 nts for the RTT yield the highest editing efficiency. Statistical significance was defined as an |log$_2$FC | > 1, and an adjusted *p*-value < 0.05. Statistical testing was performed using the Wald test provided by DESeq2, including correction for multiple testing. Results based on four biological replicates.

design and shows that mbPE is a versatile editing tool capable of introducing single point mutations, multiple simultaneous substitutions, insertions and deletions.

### Exploring the editing distance from the seed region

The strategies employed in this study guide the Cas9 nuclease moiety of mbPE^{S,pn} to cut within the seed region near the PAM motif or in the PAM as dictated by the spacer sequence, while the reverse transcriptase (RT) component of mbPE^{S,pn} introduces the desired genome

edit. This approach prevents Cas9 from retargeting the same location. The dependency on a PAM site is, however, a possible limitation of the mbPE system. For mbPE to be broadly applicable for the introduction of mutations, even in genomic regions without a nearby PAM sequence, editing should also be possible at sites distal from the PAM. To determine the maximum distance from the Cas9 cut site at which a single nucleotide substitution is possible, we designed a set of 100 pegRNAs with identical spacers. Each pegRNA contained RTT sequences to mutate the PAM (preventing retargeting) while introducing a SNP at varying distances from the PAM site (1-100 nts from the restriction site; Fig. 6A). As shown in Fig. 6B, edits were observed as far as 91 bp from the PAM site.

Several pegRNAs were individually cloned, each with a mutation that destroys the PAM to insert a silent mutation in *luc* and a substitution further from the PAM site. This showed that SNPs could be efficiently introduced at a distance of 33 base pairs from the PAM (Fig. 6C). Editing was also possible at further distances, but with increasingly reduced efficiency, similar to what was observed in the pooled screen (Fig. 6B).

### mbPE to introduce a split-*luc* tag for protein-protein interactions

Our pooled pegRNA assays indicated that DNA up to 52 bps can be inserted into the pneumococcal genome using mbPE (Fig. 4B). To test whether mbPE could be used to introduce protein tags, we considered the split luciferase tag. The use of split luciferase has emerged as a powerful approach to identify protein-protein interactions in live bacterial cells in which the bait is fused to the large luciferase component (LgBit) and the target to the small moiety (SmBit). When both parts of the split-luciferase are in close proximity, luciferase activity is restored and light is produced in the presence of furimazine-based (NanoLuc) substrate[39,40] (Fig. 7A). An outstanding question in the pneumococcal cell biology field is whether the muramidase MpgA and the bifunctional class A penicillin binding protein PBP1a are part of the so called elongasome[41-44], the machinery responsible for cell elongation. To address this question, we fused RodZ, a known component of the elongasome, as well as MpgA, to the LgBit. We designed a pegRNA to insert a sequence encoding a 2-amino acid flexible linker (SG), followed by the SmBit fragment (VTGYRLFEEIL), at the C-terminus of PBP1a (Fig. 7B). Of the 12 clones tested after selection on aTc, 1 contained the correct fusion, signifying a selection efficiency of 8% for introduction of a 39 bps-long tag, in line with the *luc*-library sequencing results showing very few clones containing insertions longer than 30 bps present in the pool (Fig. 4B). After transforming the mbPE^{S,pn}-created *pbp1A-SmBit* strain with *mpgA-LgBit* or *rodZ-LgBit*, respectively, cells were grown in microtiter plates in the presence of NanoLuc luciferase substrate and bioluminescence recorded. The *pbp1A-SmBit* strain was also transformed with *hu-LgBit* as a negative control since the highly expressed histone-like protein HU is not expected to interact with PBP1a[39,45]. As shown in Fig. 7C, we could detect a strong and reproducible signal for MpgA-LgBit and RodZ-LgBit but not for HU-LgBit, providing evidence that PBP1a, MpgA and RodZ are in close proximity in space and time. To further substantiate these findings, we constructed a double labeled strain in which PBP1a was tagged to a red fluorescent protein (mScarlet-I-opt) on its N-terminus, under the control of its native promoter, while MpgA was tagged at its N-terminus to the monomeric superfolder green fluorescent protein (GFP), also expressed from its native promoter. Strain VL7454 (mScarlet-Pbp1a, GFP-MpgA) was grown in C + Y medium at 37 °C and mid-exponentially growing cells were imaged by epifluorescence microscopy. As shown in Fig. 7D, clear co-localization between GFP-PBP1a and mScarlet-MpgA was observed, demonstrating that PBP1a and MpgA are part of the same complex. Demographs of individually GFP-tagged PBP1a and MpgA also showed similar localization dynamics across the pneumococcal cell cycle (Fig. 7D). These data are in line

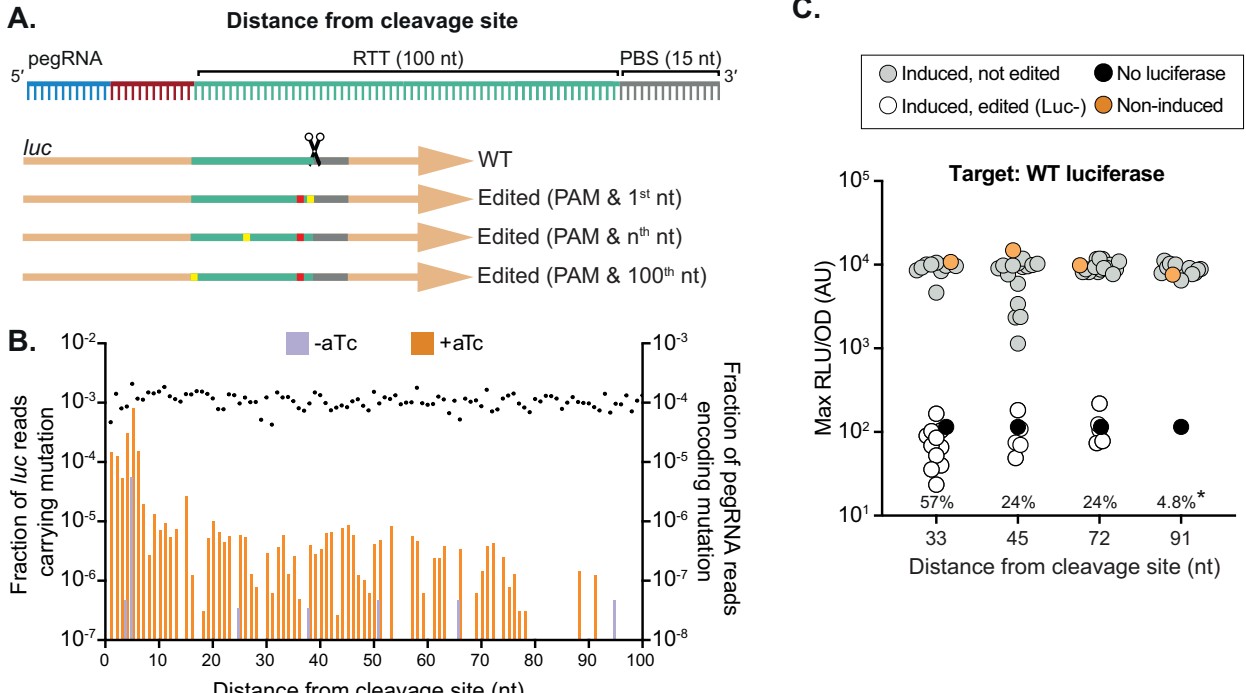

**Fig. 6 | mbPE can introduce edits at sites distal from the PAM. A** Schematic overview of the RTT design strategy used to edit bases distal from the PAM site. Every pegRNA introduces a single PAM mutation (G to C) and a second, purine to pyrimidine (or vice versa) transversion introduced at varying distances from the PAM sequence. **B** Fraction of correctly edited *luc* alleles and pegRNA read counts over the total reads in that sample of DNA extracted from cells grown in the absence (control) or presence of aTc. The average fraction of pegRNA reads for the substitutions is shown as dots on the right Y-axis. Edits are observed up to 91 bp from the PAM site. Resulting means of four biological replicates are plotted. **C** Validation of the maximum editing distance through individually cloned pegRNAs. The mutations conferred by the pegRNAs all encode the G to C PAM mutation and a second mutation at varying distances. The 33, and 45 bps substitutions from the PAM site, confer a premature stop codon, whilst the pegRNAs encoding mutation either 72 or 91 bps from the PAM site resulted in a glycine to cysteine (72 bps) or isoleucine to asparagine (91 bps) substitution ($n = 21$ each). The percentages indicate the fraction of clones that were phenotypically observed to be edited. Clones with an RLU/OD value below $3 \times 10^3$ were considered to be edited. The mutation at 91 nts (*) could only be confirmed by sequencing individual clones, as the amino acid change does not affect luciferase function. This showed an editing efficiency of 4.8 % (1 out of 21 tested colonies) with the correct genome edit.

with recent work demonstrating that both PBP1a and MpgA interact with the PBP1a-activating S protein and thus are likely part of the same protein complex[45].

## Sequential genome editing with mbPE[S.pn]

The ability to introduce several genome edits sequentially would greatly accelerate genetic engineering projects in *S. pneumoniae*. To facilitate this, we constructed a second pegRNA cloning vector with identical integration sites as plasmid pVL4133 but a different antibiotic resistance marker, conferring resistance against chloramphenicol instead of spectinomycin (pVL7227) (Fig. 7E, F). In this way, after the first genome edit has been made, the pegRNA can be replaced by transformation of the second vector containing a different pegRNA. As a proof of concept, we first introduced a 63 bps deletion in *luc* in strain VL6307 using pegRNA plasmid pVL6304, resulting in strain VL6926 (Δ63-*luc*). Next, we designed and generated pegRNA constructs that would introduce stop mutations in either the capsule gene *cps2A* or the major autolysin *lytA*. Both genes encode major pneumococcal virulence factors (Fig. 7E). By sequential transformation of strain VL6926 with plasmids pVL7230 (chloramphenicol resistance, pegRNA-Δ*cps2A*-stop) and pVL5524 (spectinomycin resistance, pegRNA-Δ*lytA*-stop), we could readily obtain strain VL7292 containing the three desired mutations (Δ63-*luc*, Δ*cps2A*-stop, Δ*lytA*-stop) (Fig. 7F). Sanger sequencing verified the correct edits in two clones of the triple mutant (Supplementary Fig. 3A), whilst whole genome PacBio sequencing of two clones of the single mutant and two clones of the triple mutant demonstrated the

absence of off-target edits or genomic rearrangements (Supplementary Fig. 3B, C) (SRA accession number PRJNA1062582).

## Discussion

The principal contribution of this work is the development of make-or-break prime editing (mbPE), a highly effective bacterial genome editing system. In contrast to traditional prime editing in which a nickase is used, mbPE utilizes a Cas9 nuclease fused to a reverse transcriptase. Combined with the high recombinogenic potential of *S. pneumoniae*[23], the edit encoded within the pegRNA that also disrupts the PAM or seed region is integrated into the genome by a mechanism that does not require RecA (Fig. 2E). It is tempting to speculate that, in absence of RecA, a single strand annealing mechanism[46] or another kind of microhomology-mediated repair process[9], such as the RecBCD (RexAB in *S. pneumoniae*[47]) or RecFOR loader complexes[48] play a role in consolidating the edit encoded by the pegRNA during mbPE. RecJ may be a key enzyme to remove the 5′ flap and promote DNA repair (Fig. 2B). Since Cas9 cannot re-target correctly edited cells, as they would lack a PAM or seed region, they are efficiently selected, while unedited clones die due to repeated Cas9-induced DNA breaks. This model for mbPE implies that editing efficiencies might be further improved by reducing the efficiency of Cas9 cleavage to allow for enough time to insztall the edit, something also observed for traditional CRISPR gene editing in bacteria[8]. Since mbPE does not require homology-dependent recombination through RecA, it might also be an efficient method in other bacteria that are less recombinogenic compared to *S. pneumoniae*.

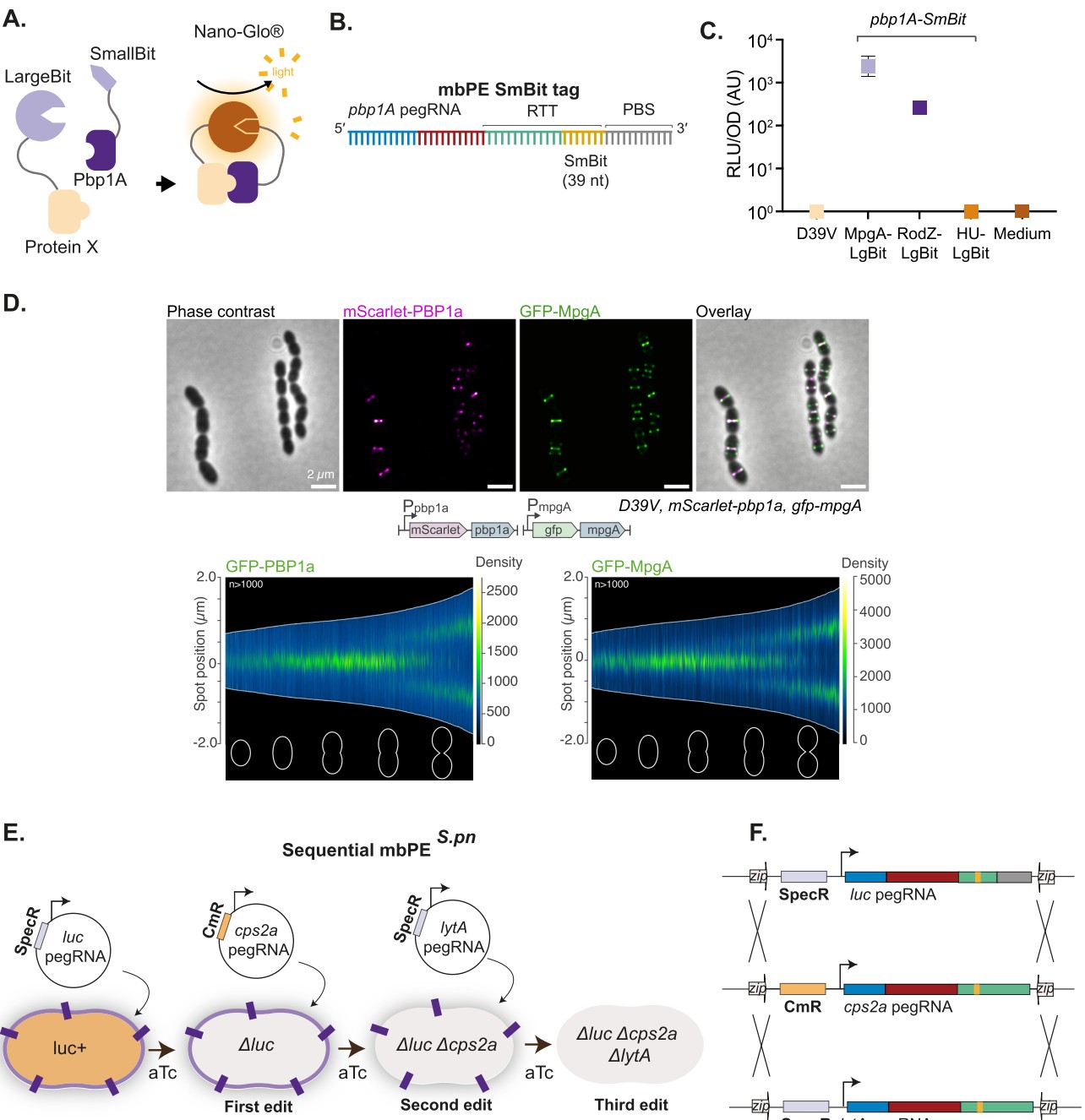

**Fig. 7 | PBP1a is in a complex with MpgA and RodZ and sequential genome editing using mbPE. A** Schematic overview of the principle of the split *luc* system. **B** Schematic overview of the pegRNA used to fuse a SmBit tag to *pbp1A* using mbPE. **C** PBP1a interacts with MpgA and RodZ. Each dot represents the average of 15 measurements of a technical replicate, with the size of the dot representing the SEM. Control strains only expressing *pbp1A-SmBit* (labeled as D39V) or *pbp1A-SmBit* together with the abundant DNA-binding protein HU-LgBit did not demonstrate bioluminescence. **D** PBP1a and MpgA co-localize in space and time. Live cell fluorescence microscopy of cells of strain VL7454 (mScarlet-PBP1a, GFP-MpgA) grown at 37 °C in C + Y medium. Scale bar: 2 mm. Demograph of exponentially growing cells of strains VL5532 (GFP-PBP1a) and VL5816 (GFP-MpgA). Representative data are shown from one biological replicate, data of at least 1,000 cells are plotted. **E**, **F** Sequential gene editing in *S. pneumoniae* using mbPE. Schematic overview of cloning regime with the two alternative pegRNA vector backbones (pVL4133, *spec*; pVL7227, *cat*) to generate a triple genome edited *S. pneumoniae* strain (Δ63-*luc*, Δ*cps2A-stop*, Δ*lytA-stop*) in a sequential manner.

Recently developed nuclease-based prime editing systems in eukaryotic cells have also shown improved selection efficiencies[19–21,49–51]. However, these systems often produce more undesired by-products, such as insertions and deletions (indels), than the intended edits[52]. Probably because *S. pneumoniae* does not, or very poorly, performs template-independent end joining (like most other bacteria[2,9,10]), we have not observed off-target indels with mbPE$^{S.pn}$ when editing efficiencies were high. Bystander mutations, if present, were rather found near the Cas9 cleavage site that are typically also found during regular PE2 prime editing[7] or within mbPE$^{S.pn}$ itself (Supplementary Fig. 2G). Long-read PacBio sequencing of triply sequential edited strains (Fig. 7, and Supplementary Fig. 3B, C) also did not show evidence of genomic rearrangements or untargeted indels.

By screening pooled libraries containing a suite of pegRNAs with different RTT and PBS lengths to introduce point mutations at a specific site (*luc*), we established that ideal PBS and RTT lengths for a

single adenine insertion in *luc* are 16 nts and 15 nts, respectively (Fig. 5). In addition, mbPE was able to make all 12 possible nucleotide substitutions as well as insertions up to 52 bps (Figs. 3, 4, Supplementary Fig. 4). As a proof of principle, we used mbPE to tag the important amoxicillin-resistance determinant PBP1A[53] with the SmBit split luciferase moiety and unequivocally showed that PBP1A is in close spatial and temporal proximity to MpgA and RodZ. Deletions up to 127 bps could also be efficiently introduced (Fig. 4C–E). By creating pegRNA vectors with various antibiotic resistance markers, sequential genome editing is a straightforward procedure and after every consecutive edit the pegRNA is simply replaced by the next (Fig. 7D, E). Together, this work demonstrates that mbPE is a flexible, scalable and efficient genome editing tool for *S. pneumoniae*.

Due to its natural competence, *S. pneumoniae* has been a cornerstone of molecular biology[54]. This feature also enabled us to optimize mbPE for genome editing. It will now be interesting to test whether mbPE is also effective in other bacteria, particularly in strains that are difficult to transform. By placing both the prime editor and the pegRNA cloning cassette on a replicative plasmid with a temperature-sensitive origin of replication, straightforward genome editing in hard-to-transform strains might be enabled. Another advantage of *S. pneumoniae* is that its genome, which is 2.04 Mb in size in case of strain D39V[35], contains 145,892 *S. pyogenes* Cas9 PAM sites (NGG) distributed fairly evenly across its sequence. This results in one PAM site every 14 base pairs on average. If a distance of 50 bp from a PAM site is considered as the maximum distance to be efficiently edited by mbPE$^{S,pn}$ (Fig. 6B), 99.845% of the D39V genome would be in range for efficient substitution editing through this system, leaving only 3158 bp unavailable. Although SNPs can be efficiently introduced at locations near a PAM site, the efficiency decreases as the distance from the PAM site increases (Fig. 6, and Supplementary Fig. 4). This is thus not a major issue for *S. pneumoniae*, but could be problematic for other bacterial species, particularly those with very low GC content. To obtain more genome editing flexibility, Cas9 variants with relaxed or different PAM requirements[55] could be used as nuclease within the mbPE. For instance, highly promising mbPE nucleases would be SpRY and SpRYc, which are engineered near-PAMless *S. pyogenes* Cas9 variants[56,57].

While the *selection* efficiency of mbPE can potentially be greater than 93%, if the mutation being introduced concerns a simple edit taking place at the PAM site (Fig. 2), the actual *editing* efficiency is typically less than 5% (Fig. 1). This low editing efficiency thus hinders multiplex genome editing to introduce multiple edits simultaneously as the chance that a cell made multiple edits before being targeted by the nuclease domain is currently smaller than the selection of suppressor mutants. Another limitation of prime editing are the restrictions in insertion as well as deletion lengths. Therefore, future make-or-break prime editors would also benefit from improvements in reverse transcriptases and pegRNA design driven by genetic screens, advances in machine learning and/or structural insights[58,59]. Removal of certain host factors, such as 3′→5′ DNA exonucleases and the mismatch repair system, may also improve editing efficiencies[18,33,34]. Future optimized mbPEs with PAMless nucleases and efficient reverse transcriptases with optimal pegRNA designs would enable multiplex editing of every base in a genome. Dual pegRNA approaches may also increase the efficiency of large genomic deletions, and facilitate the integration of larger DNA segments[60,61].

While not yet perfect, the here-described mbPE system still represents one of the most powerful and versatile bacterial genome editor currently available due to its scalability, meaning the ability to clone pools of pegRNAs to generate vast libraries of potential gene edits, and we foresee many exciting and important applications, especially since the quality (and price) of oligo pools has significantly improved in the last years. For instance, by designing a pool of pegRNAs, mbPE could be employed to perform 'all codon scanning' on single genes of interest to gain new important insights into protein function, protein interaction, or identify temperature-sensitive alleles. Another application could be to examine gene essentiality at a genome-wide scale. Since mbPE can easily introduce mutations without polar effects, large, genome-wide stop-codon or start-codon mutant libraries could be made.

The here-described mbPE system will facilitate genome-editing approaches in the significant human pathogen *S. pneumoniae* and may contribute to the discovery of new insights with regards to pneumococcal cell biology as well as lead to innovative therapeutic approaches. The described methods and approaches may serve as a roadmap to develop mbPE for other bacteria.

## Methods

### Bacterial strains, culture conditions and transformation
All strains, plasmids and primers used in this study are listed in Supplementary Data 2 and Supplementary Data 3. Pneumococcal strains in this study are derivate of *S. pneumoniae* D39V[30], unless specified otherwise, and are listed in Supplementary Data 2. Construction of strains is described in the Supplementary Information. Strains were grown in liquid semi-defined C + Y medium[55] (pH 6.8) or Columbia Agar Base (CAB) supplemented with 5% sheep blood at 37 °C. For transformation, pneumococci were grown in C + Y medium to an optical density (OD$_{595nm}$) of 0.1. Competence was activated by adding 100 ng/mL (CSP-1) (synthetic competence-stimulating peptide 1) for 12 minutes at 37 °C. The donor DNA was added to the culture and incubated 20 minutes at 30 °C. The cultures were then diluted 10 times in C + Y and incubated at 37 °C for 1.5 h. Transformants were selected in CAB containing 5% blood and the appropriate antibiotic. Re-streaked and confirmed clones were grown in C + Y to an OD$_{595nm}$ of 0.3 and stored at −80 °C in C + Y containing 17% glycerol.

*E. coli* Stbl3 (Thermofisher) was used for plasmid cloning. Strains were cultured in LB or LBA with appropriate antibiotics.

### Cloning of pegRNAs by Golden Gate Assembly
The pPEPZ_pegRNA vector was digested with Esp3I and purified from agarose gel to ensure removal of the *mCherry* fragment. The two oligonucleotides designed for the spacer, the scaffold and the extension sequences of each pegRNA were annealed in TEN buffer (10 mM Tris, 1 mM EDTA, 100 mM NaCl, pH 8) in a thermocycler at 95 °C for 5 minutes and then slowly cooled down at 25 °C. Equimolar ratios of each annealed pegRNA pieces were phosphorylated with T4 polynucleotide kinase (New England Biolabs). The annealed and phosphorylated pieces were ligated with the pPEPZ_pegRNA digested vector with T4 DNA ligase. The ligation products were transformed in *E. coli* Stbl3 and the transformants were selected on LBA plates with the appropriate antibiotic. Details about each pegRNA plasmid construction are indicated in Supplementary Information.

### Induction of PE2 and mbPE$^{S,pn}$
Strains containing the PE2 nickase prime editor or the mbPE$^{S,pn}$ prime editor were grown in C + Y to an OD$_{595nm}$ of 0.3 (3 × 10⁸). Serial dilutions were prepared in C + Y medium and plated in CAB plates containing 5% blood and a range of aTc (between 2−50 ng/mL aTc) and incubated overnight at 37 °C. A control without aTc was included to compare the colony number with and without inducer. Single colonies of plates in which ~10−50x fewer colonies were present than in the 0 aTc plates at the same dilution were picked and cultured in C + Y for further experiments. Typically, when > 100X less colonies were present on the +aTc plates compared to the -aTc plates, editing was not efficient enough compared to cutting by the nuclease and a lower aTc concentration was used for induction.

### Growth assays and luciferase activity assays
Single colonies from PE2 or mbPE induction plates were picked and pre-cultured in 200 μl of C + Y with no inducer and incubated at 37 °C.

After 2 h of incubation, 50 μL of the colony suspension were transferred to white, clear bottom 96-well plates containing 250 μL of C + Y plus 0.12 mg/mL luciferin (final luciferin concentration, 0.1 mg/mL; BioSynth). For each assay D39V, uninduced colonies and media were included as controls in duplicates. Absorbance (OD$_{595}$) and luminescence (relative luminesce units, RLU) were measured for 20 h with an interval time of 10 minutes, using a Tecan (Spark) plate reader. Growth curves were plotted using BactEXTRACT[62], normalizing the OD to 0.003. RLU/OD values were normalized against the baseline level of medium luminescence obtained for each experiment and data analyzed using GraphPad Prism.

### Nano-glo dual-luciferase assay

Strains were pre-cultured in C + Y to an OD$_{595}$ of 0.3. Luminescence (relative luminesce units, RLU) and Absorbance (OD$_{595}$) of sample triplicates supplemented with NanoGlo Live Cell Assay System (Promega) as recommended by the manufacturer, were monitored using a Tecan (Spark) plate reader during 15 minutes without intervals.

### pegRNA pool cloning

The backbone for the cloning of the pegRNA pool (pVL4134) was linearized by XbaI digestion. The linearized vector was then amplified with primers OVL7743 and OVL7744 to introduce flanking Esp3I cleavage sites. Template plasmid was removed by digestion with DpnI. The ssDNA pegRNA pool (Supplementary Data 1, Twist Bioscience) was amplified using primers OVL7966 and OVL6481 and the PCR product was excised from a 2% agarose gel using the Monarch DNA cleanup and gel extraction kit (NEB). One-step Golden Gate Assembly (GGA) was used to ligate the vector and the pegRNA pool using Esp3I (molar ratio vector insert 1:10). The GGA product was transformed in *E. coli* Stbl3 and transformants were selected on LBA plates containing 100 μg/mL spectinomycin. After overnight incubation at 37 °C, several single colonies were picked to check the cloning efficiency and the rest of the colonies were scraped-off from the plates to prepare glycerol stocks with the suspension. The plasmid pegRNA pool was purified and transformed into *S. pneumoniae* VL5155 and transformants were plated on CAB plates supplemented with 5% blood and 100 μg/mL spectinomycin and incubated overnight at 37 °C. The cells were scraped-off the plates and the bacterial suspension was stored at -80 °C in C + Y containing 17% glycerol (pool name: VL5315).

### pegRNA pool screening

To determine the best conditions for the pegRNA pool screening, a pilot mbPE induction experiment was performed. 100 μL of VL5355 were resuspended in 4 mL of C + Y containing 100 μg/mL spectinomycin, in triplicates. Starting OD$_{595}$ was 0.17 and the cells were grown to an OD$_{595}$ of 0.34. Serial dilutions of the bacterial suspension (up to 10$^{-5}$) were prepared in C + Y containing 100 μg/mL spectinomycin and plated onto 75 mL CAB + 5% blood plates with and without 50 ng/mL aTc and incubated overnight at 37 °C. The CFU were counted, and it was determined that the selection efficiency was about 10%. The dilutions to be used for the screening were chosen to be 10$^{-2}$ for the inducer condition and 10$^{-3}$ for the no inducer condition (single colonies could be still observed and the plates had a good coverage). The pegRNA pool screening was performed in quadruplicates with the same conditions previously tested in the pilot experiment. All colonies grown with and without inducer were scraped-off using C + Y + spectinomycin and stored with 17% glycerol at -80 °C. Chromosomal DNA from the pegRNA pool screening samples was extracted for Illumina library preparation using the Promega kit as described previously[39].

### Targeted *luc* modification site sequencing and differential enrichment analysis

Custom oligonucleotide primers (Supplementary Data 3) consisting of P5 or P7 Illumina sequencing adapters, Illumina sequencing barcodes (N501-504/N701-704), read 1 or read 2 sequencing primer binding sites, and portions binding to the chromosomally located *luc* were used to amplify the targeted *luc* modification site using Phanta high-fidelity polymerase (Vazyme). Amplified fragments were purified from gel and sequenced on an Illumina MiSeq, using a 2 × 300 (600 cycles) paired-end v3 sequencing kit. Illumina sequencing reads were checked for quality using FastQC (v0.11.8, available from https://www.bioinformatics.babraham.ac.uk/projects/fastqc/), and paired reads were merged using PEAR (v0.9.6, available from https://www.h-its.org/software/pear-paired-end-read-merger/). PEAR merged reads were subsequently used to count the number of occurrences of each pegRNA-encoded mutation, using 2FAST2Q (v2.5.3)[63] in counter mode with consideration for the minimal Phred quality score (all set to 20), and the mismatches allowed in upstream, downstream, and queried sequences (all 0). Counts from 2FAST2Q were imported into R, and normalized using DESeq2 (v1.40.2). DESeq2[64] was subsequently used to perform the differential enrichment analysis between the control and aTc induced samples. Statistical significance for differential presence was defined as an absolute log2FoldChange > 1, and an adjusted *p*-value < 0.05. Full results of the *luc* mutation analysis can be found in Supplementary Data 4. Note that because of the high number of unedited, wild type *luc* alleles present in the amplified pools, we were unable to accurately quantify the substitution identities and their editing efficiencies for single base pair substitutions. Instead, we quantified the abundance of the pegRNAs in presence and absence of aTc as a proxy for editing efficiency of single base pair substitutions.

### Sequencing of pegRNA and differential enrichment analysis

Custom oligonucleotide primers (Supplementary Data 3) consisting of P5 or P7 Illumina sequencing adapters, Illumina sequencing barcodes (N501-504/N701-704), read 1 or read 2 sequencing primer binding sites, and portions binding to the chromosomally integrated pegRNA expression were used to amplify the pegRNAs using Phanta polymerase. Amplified fragments were purified and sequenced using an Illumina MiniSeq instrument, with a 2 × 150 (300 cycles) paired-end mid output sequencing kit. Raw Illumina sequencing reads were checked for quality using FastQC (v0.11.8, available from https://www.bioinformatics.babraham.ac.uk/projects/fastqc/), and paired reads were merged using PEAR (v0.9.6, available from https://www.h-its.org/software/pear-paired-end-read-merger/). PEAR merged reads were subsequently used to count the number of occurrences of each pegRNA-encoded mutation, using 2FAST2Q (v2.5.3)[63] in counter mode with consideration for the minimal Phred quality score (all set to 20), and the mismatches allowed in upstream, downstream, and queried sequences (all 0). Counts from 2FAST2Q were imported into R, and normalized using DESeq2 (v1.40.2). DESeq2[64] was subsequently used to perform the differential enrichment analysis between the control and aTc induced samples. Statistical significance for differential presence was defined as an absolute log2FoldChange > 1, and an adjusted *p*-value < 0.05. Full results of the pegRNA abundance analysis can be found in Supplementary Data 5.

### Genome sequencing and analysis

Genome sequencing was performed using PacBio HiFi sequencing on a Pacbio sequel II machine. PacBio HiFi reads were assembled into circular genomes using Flye[65]. Flye-produced assemblies were rotated for *dnaA* using circlator[66]. Assemblies were assessed for structural rearrangements and large indels through minimap2[67], by mapping them to an in silico constructed reference genome containing the known mutations and insertions. Genome alignments were visualized using the dotPlotly R package. For SNPs and small indel calling, the raw reads from the PacBio sequencing were mapped to the synthetic genomes using minimap2[67], and filtered for secondary and split alignments. Resulting alignments were sorted using SAMtools, and mutations were called using bcftools, hits were

filtered according to previously used methods[68]. Coverage at each position was assessed using SAMtools[69].

## Phase contrast and fluorescence microscopy

Frozen stocks were inoculated 1:100 in C + Y medium (pH = 6.8) at 37 °C until mid-exponential growth phase ($OD_{600}$ = 0.2-0.3). Cultures were then diluted to $OD_{600}$ = 0.01 and grow at 37 °C until mid-exponential growth phase ($OD_{600}$ = 0.2-0.3). Cells were imaged by adding 0.8 µL of cell suspension onto 1.2% (w/v) PBS-agarose gel pads. Pads were then placed inside a gene frame (Thermo Fisher Scientific) and covered with a cover glass.

Microscopy images were captured using a Leica DMi8 microscope with a DFC9000 GTC-VSC04862 camera, a HC PL APO 100x/1.40 oil objective and visualized using SOLA light engine (Lumencor®). Phase contrast images were acquired using transmission light (100 ms exposure). Filter sets used on the Leica DMi8 were the following: monomeric superfolder green fluorescent protein (msfGFP) (Ex: 470/40 nm Chroma ET470/40x, BS: LP 498 Leica 11536022, Em: 520/40 nm Chroma ET520/40 m) with exposure time 700 ms, mScarlet-I-opt (Ex: Laser 550 nm Chroma ET545/30 X, BS: 595 nm Chroma 69008, Em: 635/70 nm Chroma ET635/70 m) with exposure time 700 ms. Images were acquired using LasX v.3.4.2.18368 (Leica).

## Image analysis and cell segmentation

All microscopy images were analyzed using Fiji (v1.54 g, fiji.sc). Image deconvolution was performed, when appropriate, using Huygens (v17.10.0p4, SVI) with standard settings, using 15 iterations. Demograph plots of at least 1000 cells were generated using MicrobeJ (v5.13 h)[70].

## Statistics & reproducibility

Data analysis was performed using R and Prism (Graphpad). Data shown are represented as mean of at least three replicates ± SEM if data came from one experiment with replicated measurement, and ± SD if data came from separate experiments. No statistical method was used to predetermine sample size. No data were excluded from the analyzes. The experiments were not randomized. The Investigators were not blinded to allocation during experiments and outcome assessment.

## Reporting summary

Further information on research design is available in the Nature Portfolio Reporting Summary linked to this article.

# Data availability

The sequencing data from sequencing reactions of the luciferase gene and the pegRNAs, and the PacBio sequencing data generated in this study have been deposited the Sequence Read Archive under Bio-Project accession code PRJNA1062582. Source data are provided with this paper.

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

## Acknowledgements

Work in the lab of J.W.V. is supported by the Swiss National Science Foundation (SNSF) (project grants 310030_192517, 310030_200792 and 'AntiResist' 51NF40_180541) and ERC consolidator grant 771534-PneumoCaTChER. A.B.J. is supported through a Postdoctoral Fellowship grant (TMPFP3_210202) from the SNSF, and a Robert Austrian Research Award from the International Society of Pneumonia and Pneumococcal Diseases. X.L. was supported by Shenzhen University 2035 Program for Excellent Research (86901-00000216). The funders had no role in study design, data collection and analysis, decision to publish, or preparation of the manuscript.

## Author contributions

J.W.V. wrote the paper with input from all authors. M.R.G., M.V.M., A.B.J., A.S.R., and J.B. performed the experiments. M.R.G., A.B.J., X.L., M.V.M., A.S.R., J.B., and J.W.V. designed, analyzed, and interpreted the data. All authors reviewed and approved the final version of the manuscript.

## Competing interests

J.W.V. is a scientific advisory board member at i-Seq Biotechnology. The remaining authors declare no competing interests.
