## [Transparent Peer Review file · Nature Communications]

Make-or-break prime editing for genome engineering in *Streptococcus pneumoniae*

Corresponding Author: Professor Jan-Willem Veening

Version 0:

Reviewer comments:

Reviewer #1

(Remarks to the Author)

In this work, the authors test the impact of prime editing with a cleaving Cas9 in *Streptococcus pneumoniae*, what they term make-or-break prime editing or mbPE, reporting that introducing DNA cutting greatly enhances editing efficiencies and the creation of a wide range of edits. The resulting datasets provide a clear improvement over traditional prime editing (albeit neglecting 3' exonucleases—see below) and do not require a separate repair template that can be challenging to clone, setting up the potential of mbPE to be broadly useful in bacteria. They also go into incredible depth at one target site through the use of a large pegRNA library. However, the work needs extensive improvement to accurately capture prior work in the field, contain sufficient controls to justify the conclusions, and report sufficient breadth for the field to see this approach as a useful advance in bacteria. These points are described below.

Major Comments:

1. In setting up mbPE in the abstract and introduction, the authors need to better reflect prior studies and concepts:
 - L. 20-21: the repair template for HDR and the template for reverse transcription are not equivalent, particularly in resolving how bacteria respond to dsDNA breaks.
 - L. 39-44: current understanding is that recombination happens independently of Cas9 cutting, where cutting removes unedited cells. Only when cutting is reduced (PMID = 36754958) does cutting drive editing.
 - L. 44-47: Traditional Cas9-based editing involves the same number of steps, and cloning is not normally performed in a hard-to-transform bacterium.
 - L. 58-59: Traditional PE in bacteria is improperly cast as suffering from limited selection. However, ref. 17 clearly shows that 3' exonucleases are the root cause, where silencing or deleting these genes can radically boost the editing efficiency.
 - L. 60-61: WT Cas9 has been used with PE before (refs. 33-35), but not in bacteria. This point needs to be part of the introduction lest it read as if this work is the first time such a combination has been tested.
 - L. 68-69: this claim is too far removed from what's shown in the work.
2. The work misses two important controls needed to draw basic conclusions about the functions of mbPE:
 - Uninduced conditions normally serve as the baseline. However, leaky induction could drive cutting/editing in itself. It would be important to instead introduce a non-targeting guide to at least show that it gives the same results as a targeting guide without induction.
 - Given reasonable uncertainties around how repair occurs (see comment #5), it would be important to show that the reverse transcriptase is necessary (e.g., by introducing a catalytically-inactivating mutation) and that HDR is necessary (e.g., by deleting *recA*).
3. The work is built almost entirely around evaluating editing of one site within the *luc* gene in one strain of *S. pneumoniae*. This contrasts with broader claims (e.g., title, abstract, discussion) of how mbPE could be applied in diverse bacteria and across the entire genome. When the authors did attempt editing at another site (the tag insertion) and measure the editing efficiency, the efficiency was incredibly low. To be a general tool in *S. pneumoniae*, the tag insertion and the three-step editing should be sufficient, even if the claims of what can be accomplished in this bacterium need to be pulled back. For anything to be noted about other bacteria, a different bacterium, such as *E. coli*, would need to be tested. Personally, I think extension to another bacterium would be important or the approach would only be seen as valid in *S. pneumoniae*.

4. The drop in recovered colonies with mbPE is a notable limitation of the technology, particularly for hard-to-transform bacteria noted at the beginning of the work. However, this drop is only mentioned briefly in the text and coupled with Figure S2. Instead, this drop should be integrated into Figure 2 and elsewhere and discussed when comparing the technology to other existing approaches. On this note, it would be informative to the reader how this drop compares to the drop when a standard sgRNA is used (i.e., no repair present).

5. In Figure 3C, the guide frequency in library was used to make claims around editing efficiency. I'm not convinced more frequent guides are inherently better editors or even editing at all. Instead, these could merely represent less-efficient cutters, such as due to improper folding of the pegRNA, or a pegRNA could persist because it was associated with an escape mutation (e.g., in the PE). If the authors also sequenced the luc locus, it should be possible to normalize the frequency of a given edit to the frequency of the associated pegRNA, which would provide more direct insights. Also, some validation of the results would be important, such as testing a few "good" and "bad" pegRNAs.

6. All legends provide too little information to sufficiently interpret the figures. The writing also primarily specifies conclusions rather than equipping the reader to draw their own conclusion. As some (of many) examples:

- What does each dot represent in Figures 1D, 1F, and elsewhere (if a colony, then specify so and the total number of colonies screened)?
- What is delta-hexA?
- What is D39V and how do the other labeled constructs relate to PE2(S.pn) (l. 105)?
- What is Control in Figs. 3C-D and elsewhere?
- For Fig. 3D, are the fraction of pegRNA reads in the induced and uninduced sample?
- What specific edits are being introduced in Figure 6?

7. Related to the prior point, the authors include a limit-of-detection (LOD) cut-off to separate luminescent and non-luminescent colonies. However, an LOD would imply that nothing can be measured below this cutoff, creating confusion why different values below the LOD are being reported. In addition, the LOD varies widely between 100 and 100,000, raising questions how this value was determined.

8. The authors provide two demonstrations in Figure 7 that I found intriguing but mostly preliminary. First, for Pbp1A interacting with either MpgA or RodZ, the HU control was never described. My guess is that it's a control to argue that MpgA and RodZ are in fact interacting with Pbp1A in cells, although this dataset needs to be built out to draw any sort of substantive conclusion. For instance, this could include a positive control, some confirmatory results such as a pulldown, or evaluating specific domains of MpgA and RodZ. Otherwise, Figure 7's title is not sufficiently supported. Second, the sequencing results for the three-step editing needs to be provided in the figure as well as the editing efficiency and CFU drop at each step.

9. The mechanism of repair and survival remains entirely unclear and something that should be highlighted rather than passed over. Specifically, dsDNA breaks introduced by Cas9 are normally not efficiently repaired via HDR. As the break would still be present even if the non-template strand is extended by the reverse transcriptase, the underlying repair mechanisms remain to be elucidated. I would recommend avoiding any framing around the specific repair mechanisms in the beginning or in the current version of the discussion and then tackle why repair is unclear and what could be occurring in the discussion.

Other Comments:

10. Throughout, use prime symbols rather than apostrophes when referring the nucleic-acid ends.

11. L. 104-120 and elsewhere: the authors imply that a luminescent/non-luminescent clone inherently indicates whether it is edited or unedited. However, luminescence in itself doesn't mean the cells were edited or unedited. The Sanger sequencing provides some confirmation, although only then can editing be stated.

12. Building on the prior point, no Sanger results are actually provided in Fig. S1—only the extracted sequences. The actual chromatographs need to be shown. Also, the source of the colonies needs to be specified, since labeling a sequence as edited merely describes the sequence rather than the type of colony (luminescent/non-luminescent) it came from.

13. The second instance of Sanger results in Fig. S1 should be panel C and cited accordingly in the text. Only luminescent colony was reported, and the number N's creates doubt whether the sequence is truly non-edited.

14. Figure 1C: expand to show how max RFU is measured. As it stands, it appears as if the colony itself is measured for RFU.

15. L. 137: specify which strand is nicked, since both technically are by WT Cas9.

16. L. 151-152: only 11 colonies were sequenced, and it's unclear if these were initially luminescent or non-luminescent colonies (or both). Therefore, my guess is that the 93% is probably the fraction of colonies that were luminescent, and Sanger confirmed that luminescent colonies contained the expected edit. Rewording of this statement is thus needed.

17. Line 154-156. The sentence "Note that, if desired, strains can easily be cured from the prime editing system by transforming with plasmid pPEPY to exchange the mbPE cassette with a kanamycin resistance marker" does not fit in the

context of the paragraph.

18. L. 156-158: provide a substantive analysis of the two edited clones.

19. L. 191-193: the substitutions were observed at similar frequency, although no claims can be made whether these edits were efficient or not. Instead, representative examples would need to be tested.

20. L. 211: If the authors are referring to the library screen in Figure 3, then no conclusions can be drawn around editing efficiencies without testing individual examples.

21. L. 281-282: this conclusion is an overreach to me and unnecessary.

22. L. 310: how can the 8% be related to relative guide abundance in the screen?

23. L. 330-332: as a final demonstration, the sequencing results need to be incorporated into the figure. Also, what was the editing efficiency at each step?

24. L. 361-363: considering the experiments were performed at a single site, this conclusion is too broad.

25. L. 389: the 93% only applies to one editing instance and was shown to be much lower for other edits.

26. L. 399-419: I'm not sure what scalability refers to and how mbPE is better than what already exists. I also found this paragraph looking too far downstream considering the poor editing efficiencies of introducing the protein tag and the unknown editing efficiencies for the two-step editing. If anything, an example of practical library-based editing would be needed to substantiate this level of speculation with mbPE.

Reviewer #2

(Remarks to the Author)

In this study, the authors introduce make-or-break Prime Editing (mbPE), a technique that enables precise and efficient genetic modifications in the human pathogen *Streptococcus pneumoniae*. The mbPE utilizes wild-type Cas9 nuclease with a pegRNA that targets and degrades the seed region or protospacer adjacent motif. This method leverages the tendency of most bacteria to struggle with template-independent end joining, which promotes the selective enrichment of correctly edited genome clones. The authors demonstrate that mbPE effectively introduces point mutations, deletions, and targeted insertions—such as protein tags like split luciferase—with selection efficiencies exceeding 93%.

1. How does the editing performance of mbPE compare to the CRISPR HDR approach (as detailed in <https://www.nature.com/articles/nbt.2508>) and the BacPE strategy (described in <https://www.nature.com/articles/s41467-024-45114-4>)?

2. Would utilizing dual-pegRNA strategies (<https://www.nature.com/articles/s41587-021-01133-w> or <https://www.nature.com/articles/s41592-022-01399-1>) with mbPE further enhance the insertion length to over 100 bp?

3. Is it feasible to perform simultaneous mbPE editing at multiple genomic loci (e.g., 2, 3, or more) within the same cell? Additionally, how practical is it to use mbPE in combination with the DAP array (as described in <https://www.nature.com/articles/s41467-022-30514-1>) for multiplex prime editing in bacteria?

Reviewer #3

(Remarks to the Author)

Prime editing (PE), offering the capacity for precise genome editing including base substitution, small insertions, and deletions without the need for homologous recombination or double-strand DNA breaks, has seen extensive use in genome engineering within eukaryotic cells. Despite its potential, its application in bacteria remains restricted due to comparatively low editing efficiencies. In this work, the authors have developed an enhanced mbPE tool by replacing the Cas9H840A in PE2 with a wild-type Cas9, which demonstrates efficient introduction of targeted modifications in *Streptococcus pneumoniae*. Furthermore, the authors have created a pooled mutation library and outlined the design rules for mbPE. This system has been successfully applied for the assessment of protein-protein interactions via a split-luc tag, as well as for sequential genome editing employing different antibiotic resistance markers. This advancement in the mbPE system signifies a notable contribution to facilitating prime editing applications in bacterial contexts. However, several areas in the manuscript could benefit from additional clarification or amendments:

1. The mbPE system utilizes an expression cassette (Ptet-cas9-MMLVRT) that is genomically integrated into *S. pneumoniae*, with the pegRNA constructed on a separate editing plasmid. This approach limits mbPE's application to specific strains. Could the authors consider co-integrating the Ptet-cas9-MMLVRT and pegRNA onto a single plasmid, enabling straightforward application of mbPE for genome editing in wild-type strains?

2. On lines 44-47, the discussion on the drawbacks of CRISPR/Cas9-based genome editing in bacteria might imply to readers that the described cloning steps occur within the target bacteria, whereas these steps are typically performed in *E. coli* strains such as DH5 α or Top10, and thereafter, the editing plasmid is introduced into the target bacteria. The manuscript should clarify this process to prevent misinterpretation.
3. In Figure 1B, the depicted sticky ends of the Scaffold ("CACG") and the Extension ("GCGC") sections do not fully match. This discrepancy warrants correction or clarification.
4. The editing efficiency in Figures 1D and 1F is measured via RLU/OD values of individual colonies. For a more precise evaluation of editing efficiency, employing next-generation sequencing (NGS) is recommended.
5. The results from the PacBio experiment, mentioned on lines 156-158 and 330-332, would be better visualized through figures or tables rather than described solely in textual form.
6. The manuscript mentions the acquisition of a triple mutant (Δ 63-luc, Δ cps2A-stop, Δ lytA-stop) through sequential genome editing with mbPE on lines 327 and 329. Verification of correct editing at the three target sites via Sanger sequencing would strengthen these findings.
7. In Figure S1B, the mutation site "CGC" might be erroneously labeled since it appears that "taa" was altered to "GCT," not "CGC". Additionally, the inclusion of sequencing chromatograms from Sanger sequencing would substantiate these results.

Reviewer #4

(Remarks to the Author)

Version 1:

Reviewer comments:

Reviewer #1

(Remarks to the Author)

The authors have taken substantive steps to address the comments by all three reviewers--including the addition of numerous experiments. These experiments not only provided additional controls and support for major claims but also revealed that RecA central to homologous recombination was not necessary for editing.

Accordingly, the manuscript is greatly improved. I only have one remaining and simple (yet important) comment that can be quickly resolved through a targeted change to the text.

In regards to Reviewer #1, Comment #4: the authors reasonably argue that the drop in CFUs is not an issue precisely because the culture can be expanded before applying the inducer. At the same, this drop could be critical when introducing more difficult edits (e.g., larger edits explored by the authors) or when creating a pooled library of edits. Looking closely at the text, the authors note that their approach yielded 1% - 5% surviving colonies (l. 181)--presumably in comparison to an uninduced control. However, the cited figure (2C) doesn't provide any related data. Instead, the only related data can now be found in Fig. S2B added as part of the revision, which shows a 1,000-fold drop in CFUs (or 0.1% surviving colonies). I think the percentage of surviving colonies should be revised to reflect the presented data, with S2B cited accordingly. Importantly, I don't see this drop being a deterrent to the method, although it's important information that should be available to any potential adopter of the technology.

Reviewer #2

(Remarks to the Author)

The current manuscript fails to adequately address the comments raised in the previous round of review. Below are my updated major concerns:

1. Lack of Novelty in the mbPE Approach

The manuscript gives the impression that the mbPE strategy is a novel approach, which could mislead readers. In reality, the mbPE strategy is conceptually incremental, building on previous work (PMID: 34534334) that used a Cas9 nuclease alternative to nickase Cas9 for prime editing, with the only significant modification being its application to *Streptococcus pneumoniae* instead of human cells. This change alone does not justify the claims of novelty made throughout the manuscript, particularly in the title, abstract, introduction, and conclusion.

The manuscript should be reframed to emphasize that the primary contribution is the demonstration of the applicability of a previously reported method in a new bacterial species. For example, phrases such as "we developed mbPE..." should be revised to "we demonstrate the application of prime editing with Cas9 nuclease in..." to reflect the actual contribution. The title should also be reworded to highlight the results and data rather than promoting the "make-and-break" PE as a novel development.

2. Inadequate Experimental Validation and Comparison to Existing Gene Editing Methods

The manuscript fails to provide sufficient experimental evidence to justify the claims made about the potential of mbPE. The use of a luciferase gene is not a robust demonstration of the method's broader applicability, and more compelling data are needed, especially for genomic editing at multiple loci in *Streptococcus pneumoniae* and other bacterial strains. The manuscript would benefit from additional experiments showing the versatility and efficiency of mbPE in a range of genomic contexts.

Moreover, the manuscript does not adequately benchmark mbPE against other established bacterial gene editing strategies, such as BacPE (PMID: 38280845) and CRISPR HDR (PMID: 23360965). These methods have been demonstrated to perform genome editing at multiple loci with high efficiency in various bacterial species. A detailed comparison of mbPE with these methods is essential to assess its relative advantages and limitations. Without such a comparison, the manuscript's claims of effectiveness and novelty remain unsubstantiated.

Additionally, the manuscript would benefit from expanding the scope of mbPE to include multiplex editing, at least at two loci, which is a critical feature in modern gene editing technologies. Given that CRISPR HDR (PMID: 23360965) has already demonstrated the capability for multiplex editing in *S. pneumoniae* with 75% efficiency, the failure to extend mbPE to this capability limits the potential impact of the approach.

Reviewer #3

(Remarks to the Author)

The authors have fully addressed my previous concerns.

Reviewer #4

(Remarks to the Author)

Version 2:

Reviewer comments:

Reviewer #1

(Remarks to the Author)

In this most recent round of reviews, the authors have sufficiently addressed the newest comments, including updating the title, specifying the drop in CFUs, addressing the prior work that also combined PE with WT Cas9 in animal cells, and whether more extensive experiments are warranted.

To the points raised by reviewer #2:

- The prior work (PMID = 34534334) is worth briefly citing, although it in no way undercuts the novelty or impact of the present work. For one, editors do not necessarily translate between bacterial and eukaryotic cells, and the effect of dsDNA breaks is wholly distinct. If anything, the outcomes in the prior work are distinct from those shown in the present work.
- In my view, the authors have sufficiently demonstrated their approach in different ways that matches (or even exceeds) that typically done when reporting new bacterial editing tools.

Reviewer #2

(Remarks to the Author)

I respectfully disagree with the author's arguments and maintain my previous opinions. Their reluctance to address essential concerns and further enhance the manuscript compromises its suitability for publication in Nature Communications.

We would like to thank the reviewers for their supportive comments and suggestions. We think that the changes we have made to take account of these points have significantly improved the manuscript and provide specific point-by-point responses to the reviewers' comments and other changes made to the manuscript below. We also note that we have included new fluorescence microscopy data demonstrating co-localization between PBP1a and MpgA. Consequently, the paper now also includes Jessica Burnier as author.

Reviewer #1 (Remarks to the Author):

In this work, the authors test the impact of prime editing with a cleaving Cas9 in *Streptococcus pneumoniae*, what they term make-or-break prime editing or mbPE, reporting that introducing DNA cutting greatly enhances editing efficiencies and the creation of a wide range of edits. The resulting datasets provide a clear improvement over traditional prime editing (albeit neglecting 3' exonucleases—see below) and do not require a separate repair template that can be challenging to clone, setting up the potential of mbPE to be broadly useful in bacteria. They also go into incredible depth at one target site through the use of a large pegRNA library. However, the work needs extensive improvement to accurately capture prior work in the field, contain sufficient controls to justify the conclusions, and report sufficient breadth for the field to see this approach as a useful advance in bacteria. These points are described below.

Major Comments:

1. In setting up mbPE in the abstract and introduction, the authors need to better reflect prior studies and concepts:

- L. 20-21: the repair template for HDR and the template for reverse transcription are not equivalent, particularly in resolving how bacteria respond to dsDNA breaks.
- L. 39-44: current understanding is that recombination happens independently of Cas9 cutting, where cutting removes unedited cells. Only when cutting is reduced (PMID = 36754958) does cutting drive editing.
- L. 44-47: Traditional Cas9-based editing involves the same number of steps, and cloning is not normally performed in a hard-to-transform bacterium.
- L. 58-59: Traditional PE in bacteria is improperly cast as suffering from limited selection. However, ref. 17 clearly shows that 3' exonucleases are the root cause, where silencing or deleting these genes can radically boost the editing efficiency.
- L. 60-61: WT Cas9 has been used with PE before (refs. 33-35), but not in bacteria. This point needs to be part of the introduction lest it read as if this work is the first time such a combination has been tested.
- L. 68-69: this claim is too far removed from what's shown in the work.

We have better clarified the abstract and introduction to better reflect prior studies and concepts in the revised manuscript (see document with track changes for the exact details). We thank the reviewer for pointing out the Collias et al. paper, which unfortunately completely escaped our attention and offers some good insights into how to further improve mbPE.

2. The work misses two important controls needed to draw basic conclusions about the functions of mbPE:

- Uninduced conditions normally serve as the baseline. However, leaky induction could drive cutting/editing in itself. It would be important to instead introduce a non-targeting guide to at least show that it gives the same results as a targeting guide without induction.

This is a valid point. We have included new experiments in the revised manuscript in which we have introduced a non-targeting pegRNA (a pegRNA editing *luc* in a genetic background lacking the *luc* gene) and in parallel introduced the same pegRNA that does target (a genetic background containing *luc*) and counted colonies on plates with and without aTc. As shown in new Fig. S2B, the number of colonies in absence of inducer are similar in both scenarios, demonstrating that there is not much leakiness of the mbPE system driven by the Ptet promoter. This is in line with our detailed characterization of this pneumococcal optimized tetracycline-inducible system showing it's the tightest promoter we currently have for *S. pneumoniae* while still having excellent dynamic range of induction (Sorg et al., PNAS 2020).

- Given reasonable uncertainties around how repair occurs (see comment #5), it would be important to show that the reverse transcriptase is necessary (e.g., by introducing a catalytically-inactivating mutation) and that HDR is necessary (e.g., by deleting *recA*).

We would like to highlight that the essentiality of the reverse transcriptase domain for prime editing has been demonstrated before (see for instance the seminal work by the group of David Liu). Nevertheless, the reviewer is correct that we cannot formally exclude the possibility that somehow *S. pneumoniae* is able to incorporate the correct edit in its genome from the pegRNA without reverse transcribing the RNA first before incorporating it in its genome. Therefore, we removed the reverse transcriptase domain from the mbPE construct, essentially constructing a wild-type *Ptet-cas9* strain and tested whether, in combination with a very efficient pegRNA to repair *luc*, which previously gave rise to a selection efficiency of >90%, any correctly edited cells were recovered. As shown in new Fig. 2D, of the few surviving colonies in the presence of aTc, none contained the pegRNA-encoded edit demonstrating that reverse transcription, as expected, is essential for prime editing.

To test whether *RecA*-dependent homologous recombination is necessary for make-or-break prime editing, we generated a *recA* deletion in an otherwise highly efficient mbPE *luc* strain. As shown in new Fig. 2E, all the 42 tested colonies grown on aTc in the *recA* deletion background were successfully repaired for *luc*, demonstrating that mbPE does not depend on *RecA* and likely depends on the previously proposed gap filling and ligation mechanism (Anzalone et al. 2019, Nature). This is now also highlighted in the Abstract and Discussion of the revised manuscript and suggests that homologous recombination is not a prerequisite for prime editing. This is of particularly valuable interest in developing prime editing approaches in bacteria that are poorly recombinogenic. We thank the reviewer for the excellent suggestion of this experiment.

3. The work is built almost entirely around evaluating editing of one site within the *luc* gene in one strain of *S. pneumoniae*. This contrasts with broader claims (e.g., title, abstract, discussion) of how mbPE could be applied in diverse bacteria and across the entire genome. When the authors did attempt editing at another site (the tag insertion) and measure the editing efficiency, the efficiency was incredibly low. To be a general tool in *S. pneumoniae*, the tag insertion and the three-step editing should be sufficient, even if the claims of what can be accomplished in this bacterium need to be pulled back. For anything to be noted about other bacteria, a different bacterium, such as *E. coli*, would need to be tested. Personally, I think extension to another bacterium would be important or the approach would only be seen as valid in *S. pneumoniae*.

We agree that it would be great if mbPE could be extended to other bacteria, particularly those hard to transform otherwise. However, this is very much out of the scope of the current work, which serves as a new foundational tool for the very large pneumococcal research community and serves as a roadmap to establish the method in other bacteria. We like to note that *selection efficiencies* are extremely high (>90%) for all edits that we've tried in our lab so far, as long as we stay within the guidelines here established (RTT and PBS of ~16 nts, mutating the PAM, and making substitutions). Indeed, large deletions and insertions are at this point not very efficient due to the poor *editing efficiency* of mbPE. This is now better clarified in the revised manuscript as well as some of the claims are stated in a more tempered manner.

4. The drop in recovered colonies with mbPE is a notable limitation of the technology, particularly for hard-to-transform bacteria noted at the beginning of the work. However, this drop is only mentioned briefly in the text and coupled with Figure S2. Instead, this drop should be integrated into Figure 2 and elsewhere and discussed when comparing the technology to other existing approaches. On this note, it would be informative to the reader how this drop compares to the drop when a standard sgRNA is used (i.e., no repair present).

We do not think this drop in recovered colonies is a real-world limitation as you can easily grow the strain up to high cell densities in the absence of inducer. In addition, the drop in recovered colonies strongly depends on the editing efficiency of the particular pegRNA. However, we do agree with the reviewer that it would be interesting to know the difference in using a normal sgRNA (just introducing DSBs) compared to a pegRNA (introducing edits as well as DSBs). Therefore, we have cloned a standard *luc* targeting sgRNA with a pegRNA encoding the same spacer sequence as the sgRNA. As now mentioned in the revised manuscript, we noted an increase in recovered colonies when the pegRNA was used, in line with the editing efficiencies seen with PE2 (Fig. S2B).

5. In Figure 3C, the guide frequency in library was used to make claims around editing efficiency. I'm not convinced more frequent guides are inherently better editors or even editing at all. Instead, these could merely represent less-efficient cutters, such as due to improper folding of the pegRNA, or a pegRNA could persist because it was associated with an escape mutation (e.g., in the PE). If the authors also sequenced the *luc* locus, it should be possible to normalize the frequency of a given edit to the frequency of the associated pegRNA, which would provide more direct insights. Also, some validation of the results would be important, such as testing a few "good" and "bad" pegRNAs.

As correctly pointed out by the reviewer, we cannot do the same analysis for the pool of pegRNAs introducing varying mutations around the PAM and seed region as used for Fig. 3C since the selection efficiencies will be very different depending on the level by which the mutation affects targeting/editing efficiency or any of the factors pointed out by the reviewer. Indeed, as we now show in new Fig. S4, while mutating the PAM has extremely high selection efficiencies, the further away from the PAM, the lower the selection efficiency gets. As correctly pointed out by the reviewer, different targets with different spacers would be impossible to qualitatively compare due to different editing efficiencies. We have therefore clarified the claims around editing efficiencies mentioned in the revised manuscript. We note that we have tested several good and bad pegRNAs demonstrating that in general pegRNA ratio trends are correct (see Fig. 4E and 6C). For instance, we indeed find higher selection efficiencies, as predicted by the NGS of the pegRNAs.

6. All legends provide too little information to sufficiently interpret the figures. The writing also primarily specifies conclusions rather than equipping the reader to draw their own conclusion. As some (of many) examples:

- What does each dot represent in Figures 1D, 1F, and elsewhere (if a colony, then specify so and the total number of colonies screened)?
- What is delta-hexA?
- What is D39V and how do the other labeled constructs relate to PE2(S.pn) (l. 105)?
- What is Control in Figs. 3C-D and elsewhere?
- For Fig. 3D, are the fraction of pegRNA reads in the induced and uninduced sample?
- What specific edits are being introduced in Figure 6?

We have better expanded and clarified the figure legends in the revised manuscript.

7. Related to the prior point, the authors include a limit-of-detection (LOD) cut-off to separate luminescent and non-luminescent colonies. However, an LOD would imply that nothing can be measured below this cutoff, creating confusion why different values below the LOD are being reported. In addition, the LOD varies widely between 100 and 100,000, raising questions how this value was determined.

This is a valid point as in fact the lines do not represent the limit-of-detection but is an arbitrary cut-off used to evaluation selection efficiency of edited clones. We thank the referee for pointing out the huge differences in units between some of the panels. It turns out this was plotting mistake. The figures have been updated and the legends have been clarified.

8. The authors provide two demonstrations in Figure 7 that I found intriguing but mostly preliminary. First, for Pbp1A interacting with either MpgA or RodZ, the HU control was never described. My guess is that it's a control to argue that MpgA and RodZ are in fact interacting with Pbp1A in cells, although this dataset needs to be built out to draw any sort of substantive conclusion. For instance, this could include a positive control, some confirmatory results such as a pulldown, or evaluating specific domains of MpgA and RodZ. Otherwise, Figure 7's title is not sufficiently supported. Second, the sequencing results for the three-step editing needs to be provided in the figure as well as the editing efficiency and CFU drop at each step.

Indeed, the HU control is a very good control for these kinds of assays as HU is highly expressed and the HU-HU interaction is very strong (Gallay et al., Nature Microbiol. 2021). Thus, the absence of any signal between Pbp1A-SmBit with HU-LgBit indicates that Pbp1A does not just interact with everything. We also agree that only showing split-luc data is not sufficient to draw the conclusion that Pbp1A and MpgA are part of the same complex. Therefore, we have now generated a double labeled strain in which Pbp1A is fused to GFP and MpgA is fused to mScarlet-I. Live cell fluorescence microscopy nicely confirms the split-luc data showing that the two protein co-localize in space and time. To quantify the localization patterns, we also constructed individual GFP fusions to PBP1a and MpgA and performed cell cycle kymographs analysis. This also corroborates that they have a similar localization pattern during the cell cycle. This new data is now presented in new Figure 7D.

We have performed a more thorough sequence analysis of the strains generated by sequential mbPE. These analyses are now shown in new Fig. S3 and clearly show the absence of any genomic rearrangements. Sanger sequencing profiles confirm the WGS by illumina that the three genome edits have been precisely introduced as encoded by the pegRNAs.

9. The mechanism of repair and survival remains entirely unclear and something that should be highlighted rather than passed over. Specifically, dsDNA breaks introduced by Cas9 are normally not efficiently repaired via HDR. As the break would still be present even if the non-template strand is extended by the reverse transcriptase, the underlying repair mechanisms remain to be elucidated. I would recommend avoiding any framing around the specific repair mechanisms in the beginning or in the current version of the discussion and then tackle why repair is unclear and what could be occurring in the discussion.

We have rephrased statements concerning specific repair mechanisms potentially involved in mbPE. We do note that the new data with a *recA* mutant highlights that RecA-dependent homologous recombination is not required for efficient make-or-break prime editing (new Fig. 2E). It is thus likely that, as suggested by the referee, only the cells in which the edit has been made before the Cas9 moiety was able to cut the DNA, will survive. This has now been better discussed in the revised manuscript.

Other Comments:

10. Throughout, use prime symbols rather than apostrophes when referring the nucleic-acid ends.

Done.

11. L. 104-120 and elsewhere: the authors imply that a luminescent/non-luminescent clone inherently indicates whether it is edited or unedited. However, luminescence in itself doesn't mean the cells were edited or unedited. The Sanger sequencing provides some confirmation, although only then can editing be stated.

The referee is correct, and we have included more Sanger sequencing results in the supplemental information, demonstrating a 1:1 correlation between absence/production of light and correct editing. The text has been clarified throughout in the revised manuscript.

12. Building on the prior point, no Sanger results are actually provided in Fig. S1—only the extracted sequences. The actual chromatographs need to be shown. Also, the source of the colonies needs to be specified, since labeling a sequence as edited merely describes the sequence rather than the type of colony (luminescent/non-luminescent) it came from.

This was an unfortunate omission, and the chromatographs have now been included. In addition, more colonies have been sequenced and the source of the colonies has been clarified.

13. The second instance of Sanger results in Fig. S1 should be panel C and cited accordingly in the text. Only luminescent colony was reported, and the number N's creates doubt whether the sequence is truly non-edited.

See above.

14. Figure 1C: expand to show how max RFU is measured. As it stands, it appears as if the colony itself is measured for RFU.

The figure legend has been clarified.

15. L. 137: specify which strand is nicked, since both technically are by WT Cas9.

Done.

16. L. 151-152: only 11 colonies were sequenced, and it's unclear if these were initially luminescent or non-luminescent colonies (or both). Therefore, my guess is that the 93% is probably the fraction of colonies that were luminescent, and Sanger confirmed that luminescent colonies contained the expected edit. Rewording of this statement is thus needed.

The referee is correct, and the statement has been reworded to better reflect this.

17. Line 154-156. The sentence "Note that, if desired, strains can easily be cured from the prime editing system by transforming with plasmid pPEPY to exchange the mbPE cassette with a kanamycin resistance marker" does not fit in the context of the paragraph.

For clarity, this sentence has been removed in the revised manuscript.

18. L. 156-158: provide a substantive analysis of the two edited clones.

See new Fig. S3.

19. L. 191-193: the substitutions were observed at similar frequency, although no claims can be made whether these edits were efficient or not. Instead, representative examples would need to be tested.

Claims concerning efficiency have been removed.

20. L. 211: If the authors are referring to the library screen in Figure 3, then no conclusions can be drawn around editing efficiencies without testing individual examples.

See above.

21. L. 281-282: this conclusion is an overreach to me and unnecessary.

The sentence has been removed.

22. L. 310: how can the 8% be related to relative guide abundance in the screen?

This has been clarified in the revised manuscript: Of the 12 clones tested after selection on aTc, 1 contained the correct fusion, signifying a selection efficiency of 8% for introduction of a 39 bps-long tag, in line with the luc-library sequencing results showing very few clones containing insertions longer than 30 bps present in the pool (Fig. 4B).

23. L. 330-332: as a final demonstration, the sequencing results need to be incorporated into the figure. Also, what was the editing efficiency at each step?

A more detailed analysis of the PacBio genome sequences have been included, as well as additional Sanger sequencing profiles (new Fig. S3). The selection efficiency was nearly 100% at each step. Note that only 3 colonies from aTc plates per mutant were selected and, in each case, contained the desired edit. The system for such deletions/substitutions is so efficient that a researcher just needs to pick a few colonies.

24. L. 361-363: considering the experiments were performed at a single site, this conclusion is too broad.

This has been rephrased.

25. L. 389: the 93% only applies to one editing instance and was shown to be much lower for other edits.

This has been rephrased and better clarified.

26. L. 399-419: I'm not sure what scalability refers to and how mbPE is better than what already exists. I also found this paragraph looking too far downstream considering the poor editing efficiencies of introducing the protein tag and the unknown editing efficiencies for the two-step editing. If anything, an example of practical library-based editing would be needed to substantiate this level of speculation with mbPE.

We have removed the more far reaching potential future outlook of the technology and better clarified by what we mean by scalability.

Reviewer #2 (Remarks to the Author):

In this study, the authors introduce make-or-break Prime Editing (mbPE), a technique that enables precise and efficient genetic modifications in the human pathogen *Streptococcus pneumoniae*. The mbPE utilizes wild-type Cas9 nuclease with a pegRNA that targets and degrades the seed region or protospacer adjacent motif. This method leverages the tendency of most bacteria to struggle with template-independent end joining, which promotes the selective enrichment of correctly edited genome clones. The authors demonstrate that mbPE effectively introduces point mutations, deletions, and targeted insertions—such as protein tags like split luciferase—with selection efficiencies exceeding 93%.

1. How does the editing performance of mbPE compare to the CRISPR HDR approach (as detailed in <https://www.nature.com/articles/nbt.2508>) and the BacPE strategy (described in <https://www.nature.com/articles/s41467-024-45114-4>)?

mbPE is very different from the ‘standard’ CRISPR HDR approach since the standard approach still needs a template to introduce the edit, besides the sgRNA. While in mbPE, both the sgRNA and the edit are incorporated in the same pegRNA, making mbPE a much more scalable and less laborious method.

The BacPE strategy recently reported for *E. coli* is more similar to mbPE but uses the traditional nicking prime editor system. By mutating several host factors, such as three 3'→5' DNA exonucleases, high editing efficiencies can be obtained. mbPE stands out as the host does not need to be modified to obtain high selection efficiency. However, the BacPE paper does provide new insights that can be used to also optimize prime editing efficiency in *S. pneumoniae* that may in the future allow for, for instance, multiplex mbPE. This is now better discussed in the revised manuscript.

2. Would utilizing dual-pegRNA strategies (<https://www.nature.com/articles/s41587-021-01133-w> or <https://www.nature.com/articles/s41592-022-01399-1>) with mbPE further enhance the insertion length to over 100 bp?

This is an excellent suggestion and indeed combining such dual pegRNA approaches with mbPE might further enhance the insertion length. This has now been mentioned in the Discussion and the papers have been cited.

3. Is it feasible to perform simultaneous mbPE editing at multiple genomic loci (e.g., 2, 3, or more) within the same cell? Additionally, how practical is it to use mbPE in combination with the DAP array (as described in <https://www.nature.com/articles/s41467-022-30514-1>) for multiplex prime editing in bacteria?

We have tried multiplex mbPE editing at multiple genomic loci, but at the moment, with the current system, the editing efficiencies are too low and only rarely do we pick up correct clones. Imagine if each edit occurs at 2% efficiency, then the chance of having a cell that made both edits at the same time is 0.04%. The chance that this cell while being edited hasn't yet been killed by the nuclease activity at one of the 2 sites is very rare and likely smaller than the selection of a suppressor mutant hence why multiplex editing currently does not yet work. With the current progress made in the field related to pegRNA design, new prime editors and host engineering, multiplex prime editing certainly will become possible in the near future. This is now better explained in the revised Discussion:

“While the *selection* efficiency of mbPE can potentially be greater than 93%, if the mutation being introduced concerns a simple edit taking place at the PAM site (Fig. 2), the actual *editing* efficiency is typically less than 5% (Fig. 1). This low editing efficiency thus hinders multiplex genome editing to introduce multiple edits simultaneously as the chance that a cell made multiple edits before being targeted by the nuclease domain is currently smaller than the selection of suppressor mutants. Another limitation of prime editing are the restrictions in insertion as well as deletion lengths. Therefore, future make-or-break prime editors would also benefit from improvements in reverse transcriptases and pegRNA design driven by genetic screens, advances in machine learning and/or structural insights^{55,56}. Removal of certain host factors, such as 3'→5' DNA exonucleases and the mismatch repair system, may also improve editing efficiencies^{18,33,34}. Future optimized mbPEs with PAMless nucleases and efficient reverse transcriptases with optimal pegRNA designs would enable multiplex editing of every base in a genome. Dual pegRNA approaches may also increase the efficiency of large genomic deletions, and facilitate the integration of larger DNA segments^{57,58}.”

Reviewer #3 (Remarks to the Author):

Prime editing (PE), offering the capacity for precise genome editing including base substitution, small insertions, and deletions without the need for homologous recombination or double-strand DNA breaks, has seen extensive use in genome engineering within eukaryotic cells. Despite its potential, its application in bacteria remains restricted due to comparatively low editing efficiencies. In this work, the authors have developed an enhanced mbPE tool by replacing the Cas9H840A in PE2 with a wild-type Cas9, which demonstrates efficient introduction of targeted modifications in *Streptococcus pneumoniae*. Furthermore, the authors have created a pooled mutation library and outlined the design rules for mbPE. This system has been successfully applied for the assessment of protein-protein interactions via a split-luc tag, as well as for sequential genome editing employing different antibiotic resistance markers. This advancement in the mbPE system signifies a notable contribution to facilitating prime editing applications in bacterial contexts. However, several areas in the manuscript could benefit from additional clarification or amendments:

1. The mbPE system utilizes an expression cassette (Ptet-cas9-MMLVRT) that is genomically integrated into *S. pneumoniae*, with the pegRNA constructed on a separate editing plasmid. This approach limits mbPE's application to specific strains. Could the authors consider co-integrating the Ptet-cas9-MMLVRT and pegRNA onto a single plasmid, enabling straightforward application of mbPE for genome editing in wild-type strains?

Good suggestion! Since we can harness natural competence in *Streptococcus pneumoniae* to, relatively easily, integrate DNA stably into its genome, we first went for this route since we have noticed that *E. coli/S. pneumoniae* shuttle vectors containing *cas9* tend to be unstable. By decoupling the prime editor from the pegRNA cloning vector, we did not have to worry about plasmid stability issues. However, for future work, and especially for other bacteria where chromosomal integration is less straightforward, it would be an excellent idea to create a replicative plasmid, for instance with a thermolabile origin of replication, containing both the mbPE and the pegRNA cloning cassette. This has now been discussed in the revised manuscript.

2. On lines 44-47, the discussion on the drawbacks of CRISPR/Cas9-based genome editing in bacteria might imply to readers that the described cloning steps occur within the target bacteria, whereas these steps are typically performed in *E. coli* strains such as DH5 α or Top10, and thereafter, the editing plasmid is introduced into the target bacteria. The manuscript should clarify this process to prevent misinterpretation.

Good point, and this has now been clarified (see also reply to Reviewer #1).

3. In Figure 1B, the depicted sticky ends of the Scaffold ("CACG") and the Extension ("GCGC") sections do not fully match. This discrepancy warrants correction or clarification.

Thanks for pointing out this mistake. The figure has been corrected. It indeed should be CACG for the Scaffold and GTGC for the Extension.

4. The editing efficiency in Figures 1D and 1F is measured via RLU/OD values of individual colonies. For a more precise evaluation of editing efficiency, employing next-generation sequencing (NGS) is recommended.

The reviewer is right that performing NGS on each edit would be the preferred method to quantitatively establish editing efficiency. We opted to go for a smaller scale approach for these initial experiments (note that for the mbPE we did use the power of NGS) in which we either introduced a stop mutation or repaired a stop mutation in the firefly luciferase gene. By performing Sanger sequencings of selected clones, we show that the production of bioluminescence is in fact a very accurate proxy for actual DNA editing. In these *luc* approaches we are of course limited by number of individual colonies tested, and the statistical power is much greater by doing NGS on thousands of clones. Nevertheless, this also works, and we have now included more Sanger traces to bring across this point in the revised manuscript and have better mentioned the limitations of the *luc* approach.

5. The results from the PacBio experiment, mentioned on lines 156-158 and 330-332, would be better visualized through figures or tables rather than described solely in textual form.

We have now included a deeper analysis of the PacBio genome sequences as new supplemental figure 3.

6. The manuscript mentions the acquisition of a triple mutant ($\Delta 63$ -luc, Δ cps2A-stop, Δ lytA-stop) through sequential genome editing with mbPE on lines 327 and 329. Verification of correct editing at the three target sites via Sanger sequencing would strengthen these findings.

Besides performing the entire genome sequence at more than 200-fold coverage demonstrating the presence of the desired edits and lack of off-targets or genomic rearrangements, we have now also included Sanger sequencing profiles demonstrating correct editing at these three target sites (new Fig. S3).

7. In Figure S1B, the mutation site “CGC” might be erroneously labeled since it appears that “taa” was altered to “GCT,” not “CGC”. Additionally, the inclusion of sequencing chromatograms from Sanger sequencing would substantiate these results.

Well spotted! This has been corrected in the revised manuscript (the wrong three bases were highlighted).

Reviewer #4 (Remarks to the Author):

We would like to thank the reviewers for their supportive comments and suggestions and have now responded to the final outstanding questions in this second revised manuscript. We provide specific point-by-point responses to the reviewers' comments and other changes made to the manuscript below.

REVIEWER COMMENTS

Reviewer #1 (Remarks to the Author):

The authors have taken substantive steps to address the comments by all three reviewers--including the addition of numerous experiments. These experiments not only provided additional controls and support for major claims but also revealed that RecA central to homologous recombination was not necessary for editing.

Accordingly, the manuscript is greatly improved. I only have one remaining and simple (yet important) comment that can be quickly resolved through a targeted change to the text.

In regards to Reviewer #1, Comment #4: the authors reasonably argue that the drop in CFUs is not an issue precisely because the culture can be expanded before applying the inducer. At the same, this drop could be critical when introducing more difficult edits (e.g., larger edits explored by the authors) or when creating a pooled library of edits. Looking closely at the text, the authors note that their approach yielded 1% - 5% surviving colonies (l. 181)--presumably in comparison to an uninduced control. However, the cited figure (2C) doesn't provide any related data. Instead, the only related data can now be found in Fig. S2B added as part of the revision, which shows a 1,000-fold drop in CFUs (or 0.1% surviving colonies). I think the percentage of surviving colonies should be revised to reflect the presented data, with S2B cited accordingly. Importantly, I don't see this drop being a deterrent to the method, although it's important information that should be available to any potential adopter of the technology.

We thank the reviewer for their supportive comments and pointing out this discrepancy. Indeed, we have now updated the number to 0.1% surviving colonies in the main text that reflects the data shown in Fig. S2B.

Reviewer #2 (Remarks to the Author):

The current manuscript fails to adequately address the comments raised in the previous round of review. Below are my updated major concerns:

1. Lack of Novelty in the mbPE Approach

The manuscript gives the impression that the mbPE strategy is a novel approach, which could mislead readers. In reality, the mbPE strategy is conceptually incremental, building on previous work (PMID: 34534334) that used a Cas9 nuclease alternative to nickase Cas9 for prime editing, with the only significant modification being its application to *Streptococcus pneumoniae* instead of human cells. This change alone does not justify the claims of novelty made throughout the manuscript, particularly in the title, abstract, introduction, and conclusion.

The manuscript should be reframed to emphasize that the primary contribution is the demonstration of the applicability of a previously reported method in a new bacterial species. For example, phrases such as "we developed mbPE..." should be revised to "we demonstrate the application of prime editing with Cas9 nuclease in..." to reflect the actual contribution. The title should also be reworded to highlight the results and data rather than promoting the "make-and-break" PE as a novel development.

To better specify the main achievements of this work, we have reformulated the title and it now reads: Make-or-break prime editing for genome engineering in *Streptococcus pneumoniae*. This makes our claims less broad and

more specific. We would like to point out that the pneumococcal research field is a huge field on its own so this paper will be very impactful for that community and is certainly not incremental. In addition, the pegRNA design rules for mbPE as well as the approaches taken here will serve as a great starting point for development of mbPE in other organisms. This work is the first to show that a nuclease can be used in combination with a reverse transcriptase to perform prime editing in a bacterium. Therefore, we feel it is warranted to coin a specific term for this kind of bacterial genome editing.

2. Inadequate Experimental Validation and Comparison to Existing Gene Editing Methods

The manuscript fails to provide sufficient experimental evidence to justify the claims made about the potential of mbPE. The use of a luciferase gene is not a robust demonstration of the method's broader applicability, and more compelling data are needed, especially for genomic editing at multiple loci in *Streptococcus pneumoniae* and other bacterial strains. The manuscript would benefit from additional experiments showing the versatility and efficiency of mbPE in a range of genomic contexts.

Moreover, the manuscript does not adequately benchmark mbPE against other established bacterial gene editing strategies, such as BacPE (PMID: 38280845) and CRISPR HDR (PMID: 23360965). These methods have been demonstrated to perform genome editing at multiple loci with high efficiency in various bacterial species. A detailed comparison of mbPE with these methods is essential to assess its relative advantages and limitations. Without such a comparison, the manuscript's claims of effectiveness and novelty remain unsubstantiated.

Additionally, the manuscript would benefit from expanding the scope of mbPE to include multiplex editing, at least at two loci, which is a critical feature in modern gene editing technologies. Given that CRISPR HDR (PMID: 23360965) has already demonstrated the capability for multiplex editing in *S. pneumoniae* with 75% efficiency, the failure to extend mbPE to this capability limits the potential impact of the approach.

We do not agree with these comments. Indeed, we have shown that we are able to engineer the luciferase gene both by performing luciferase assays combined with traditional Sanger sequencing, as well as using pooled pegRNA libraries coupled to NGS. However, not only on a single position, but at several positions within the *luc* gene were edits successfully made. In addition, we have demonstrated we can successfully edit the *pbp1A* gene (insert a split-*luc* tag) as well as mutate the *cps2A* and *lytA* genes using mbPE.

We have benchmarked mbPE against traditional PE and show that it has superior effectiveness in selecting correctly edited clones. BacPE is not a good comparison with mbPE as in BacPE the host strain first needs to be mutated for several 3'→5' DNA exonuclease whereas in mbPE no such host mutations are required. CRISPR HDR is the standard way of CRISPR-based genome editing, which we already reported on previously (PMID: 33397704) and also does not make sense to compare with mbPE as a separate HDR template needs to be provided together with an sgRNA, whereas mbPE only requires a single pegRNA that both encodes the targeting and the edit.

Reviewer #3 (Remarks to the Author):

The authors have fully addressed my previous concerns.

Reviewer #4 (Remarks to the Author):

We were pleased to see that all referees besides referee 2 were completely satisfied with our revised paper. It was very good to see that referee #1 agrees with us that the demands made by referee #2 are far out of the scope of our paper. We provide specific point-by-point responses to the reviewers' comments below.

REVIEWERS' COMMENTS

Reviewer #1 (Remarks to the Author):

In this most recent round of reviews, the authors have sufficiently addressed the newest comments, including updating the title, specifying the drop in CFUs, addressing the prior work that also combined PE with WT Cas9 in animal cells, and whether more extensive experiments are warranted.

We thank the reviewer for their supportive comments.

To the points raised by reviewer #2:

- The prior work (PMID = 34534334) is worth briefly citing, although it in no way undercuts the novelty or impact of the present work. For one, editors do not necessarily translate between bacterial and eukaryotic cells, and the effect of dsDNA breaks is wholly distinct. If anything, the outcomes in the prior work are distinct from those shown in the present work.

- In my view, the authors have sufficiently demonstrated their approach in different ways that matches (or even exceeds) that typically done when reporting new bacterial editing tools.

We thank the reviewer for their supportive comments.

Reviewer #2 (Remarks to the Author):

I respectfully disagree with the author's arguments and maintain my previous opinions. Their reluctance to address essential concerns and further enhance the manuscript compromises its suitability for publication in Nature Communications.

See our previous responses as well as the response of Reviewer #1.